

# Creative computing with Landlab: an open-source toolkit for building, coupling, and exploring two-dimensional numerical models of Earth-surface dynamics

Daniel E. J. Hobley[1,2,3], Jordan M. Adams[4], Sai Siddhartha Nudurupati[5], Eric W. H. Hutton[6], Nicole M. Gasparini[4], Erkan Istanbulluoglu[5], Gregory E. Tucker[1,2]

[1]Cooperative Institute for Research in Environmental Sciences (CIRES), University of Colorado, Boulder, USA
[2]Department of Geological Sciences, University of Colorado, Boulder, USA
[3]School of Earth and Ocean Sciences, Cardiff University, Cardiff, UK
[4]Department of Earth and Environmental Sciences, Tulane University, New Orleans, USA
[5]Department of Civil and Environmental Engineering, University of Washington, Seattle, USA
[6]Community Surface Dynamics Modeling System (CSDMS), University of Colorado, Boulder, USA

*Correspondence to*: Daniel E. J. Hobley (daniel.hobley@colorado.edu)

**Abstract.** The ability to model surface processes and to couple them to both subsurface and atmospheric regimes has proven invaluable to research in the Earth and planetary sciences. However, creating a new model typically demands a very large investment of time, and modifying an existing model to address a new problem typically means the new work is constrained to its detriment by model adaptations for a different problem. Landlab is an open-source software framework explicitly designed to accelerate the development of new process models by providing: 1. a set of tools and existing grid structures – including both regular and irregular grids – to make it faster and easier to develop new process components, or numerical implementations of physical processes; 2. a suite of stable, modular, and interoperable process components that can be combined to create an integrated model; and 3. a set of tools for data input, output, manipulation, and visualization. A set of example models built with these components is also provided. Landlab's structure makes it ideal not only for fully developed modelling applications, but also for model prototyping and classroom use. Because of its modular nature, it can also act as a platform for model intercomparison and epistemic uncertainty and sensitivity analyses. Landlab exposes a standardized model interoperability interface, and is able to couple to third party models and software. Landlab also offers tools to allow the creation of cellular automata, and allows native coupling of such models to more traditional continuous differential equation-based modules. We illustrate the principles of component coupling in Landlab using a model of landform evolution, a cellular ecohydrologic model, and a flood-wave routing model.





### 1 Introduction and motivation

Across a wide array of fields, researchers use numerical models to study processes that operate on and across the Earth's land surface and shallow subsurface. Science and engineering applications of these models of surface dynamics range from short-term flood forecasting (e.g., Horritt and Bates, 2002) to simulating the evolution of Earth's landscape

over geologic epochs (e.g., Tucker and Hancock, 2010). Models may focus on a theoretical understanding of processes and their interaction, on management or engineering applications, or on predicting environmental responses to natural or human-made perturbations. Although the processes and temporal and spatial scales vary widely, the software behind these models is often quite similar. For example, most Earth surface dynamics models manage data structures and algorithms to represent a terrain surface and its connectivity, and many include solution algorithms to compute flows of mass (such as ice, liquid

water, sediment, or chemical nutrients) across terrain (Slingerland and Kump, 2011) (Fig. 1).

However, scientists who want to use an Earth surface model often build their own unique model from the ground up, re-coding the basic building blocks of their model rather than taking advantage of codes that have already been written (Adams et al., 2014; Katz et al., 2015; Overeem et al., 2013). Whereas the end result may be novel software programs,

many person-hours are lost rewriting existing code, and the resulting software is often idiosyncratic, *ad hoc*, undocumented, and unable to interact with other software programs both in the same scientific community and beyond. In particular, models are often initially written to solve a very specific problem, rather than to provide a flexible and reliable platform for solving a general class of problems (Easterbrook, 2014). A result is that software development often acts as a bottleneck to progress, with frequent duplication of effort as research groups struggle to adapt existing software or develop new code from the

ground up as each new research problem emerges.

The Landlab modelling framework described here seeks to mitigate these redundancies and lost opportunities and simultaneously lower the bar for entry into numerical modelling. The approach is to create a user- and developer-friendly modelling environment that provides scientists with the fundamental building blocks needed for modelling surface dynamics

on the Earth, and potentially beyond. The framework takes advantage of the fact that nearly all surface-dynamics models share a set of common software elements, despite the wide range of processes and scales that they encompass (Peckham et al., 2013; Slingerland and Kump, 2011). Providing these elements in the context of the popular scientific programming language Python, and with strong user support and community engagement, would contribute to accelerating progress in the diverse sciences of the Earth's surface.

From the user's perspective, Landlab enables the following:

1. Rapid, easy creation of a number of distinct geometric *grids*, with all the connectivity between various elements already defined, and the ability to create two-dimensional data fields across a given grid;



2. Functions to operate on the values defined on such a grid, enabling the solution of time-dependent numerical algorithms across them (e.g., differential equations, cellular automata);

3. A mechanism for the control of boundary conditions across a grid;

4. Encapsulation of conceptual models for individual Earth-surface processes into reusable *components*, with a standard interface that allows operation across Landlab grids;

5. The ability to build a multi-process model by combining together components;

6. The ability to quickly and efficiently build new components, and to couple them with those components already in the library.

7. A straightforward and standardized input and output interface, including the ability to import from and export to common spatially distributed data formats such as NetCDF and ESRI ASCII, and a plotting module. This interface also enables coupling to third party models and software.

## 2 Approach

### 2.1 Guiding design principles

The design principles for Landlab have been guided both by our observations of current software design practices in the surface-system modelling community, and by white papers issued by existing organisations both within this community (Adams et al., 2014; Overeem et al., 2013; Peckham et al., 2013) and in the scientific software design community more widely (Becker et al., 2015; Chue Hong, 2014; Katz et al., 2015; NSF, 2012). Our key observations are that:

1. Many models exist that simulate Earth surface processes, and many of these share a very similar underpinning in terms of the basics of grid construction and the suite of simulated processes. This set of models represents significant past duplicative effort in the surface process modelling community. Although the reasons for duplication are likely multiple and vary from group to group, we note that we are unaware of previous efforts to advertise a flexible, open-source programming framework.

2. Orphaned or unmaintained codes are common in the community, having been built for a single purpose and then set aside.

3. Although standardized frameworks for model interoperability are now in place (such as the framework designed and maintained by the Community Surface Dynamics Modelling System (CSDMS) group (Hutton et al., 2014; Overeem et al., 2013; Peckham et al., 2013)), many models are not compatible with these standards. We hypothesize this is largely due to the effort required by the original programmer to modify legacy code – which in many cases was written before the standards were established – to meet these new interoperability criteria.



4. Existing model software tends to have a high bar to entry. Many models are written in compiled languages, such as Fortran, C, and C++ (examples from the geomorphology and sedimentary stratigraphy communities include CHILD (Tucker et al., 2001b), Sedflux (Hutton and Syvitski, 2008), MARSSIM (Howard, 2007), Fastscape (Braun and Willett, 2013), DAC (Goren et al., 2014), SIBERIA (Willgoose et al., 1991a; 1991b)). This requires the

prospective user be fluent in these languages before the code can be modified, or in many cases, even used efficiently. Because many legacy codes were not designed to be shared amongst the community, documentation, both in-line and external, tends to be idiosyncratic at best and missing at worst.

5. In several instances, scientific software with a broad user base exists, but remains closed source. This includes both tools for data analysis (e.g., ArcMap, Matlab, Topotoolbox2 (Schwanghart and Scherler, 2014), TecDEM

(Shahzad and Gloaguen, 2011)) and in some cases the modelling software itself (e.g., FLAC (Itasca, 2000), Dionisos (Granjeon and Joseph, 1999)). Where software has to be purchased, this presents obvious barriers to wide uptake of modelling approaches using these tools in terms of financial cost for the user. More importantly, all closed source software also presents significant barriers to code assessment in peer review and to reproducibility of the work (Crick et al., 2014; Katz et al., 2015).

These observations lead us to a set of key design principles that have governed our development of Landlab:

A. Landlab should be a community resource, and thus fully **open source**.

B. Landlab should provide a development environment that is **flexible, extensible, and highly reusable**.

C. Landlab should be written in a language that allows **rapid development** of new code.

D. Landlab should be fully compliant with the CSDMS **model interoperability standards** (Peckham et al., 2013) from the ground up, and this compliance should be built into the low-level development framework itself. Thus, for example, components written in Landlab will be automatically compliant with these standards.

E. Landlab should have a **low bar to entry**, and be thoroughly **documented**. Tutorials should be present. It should be possible for a beginner to use Landlab without a full grasp of the underlying model architecture, in a "plug and

play" fashion.

F. Landlab's code needs to be **sustainable**, as detailed below.

**2.2 Low level design choices**

In turn, these guiding design principles directed early decisions in terms of Landlab's coding language, architecture,

and distribution.





**Open source availability**. Landlab is licensed under the MIT free software license, an approved license of the Open Source Initiative. This license allows a user to deal in the software without restriction, including without limitation the rights to use, copy, modify, merge, publish, distribute, sublicense, and/or sell copies of the software. The source code and associated files are maintained in a git version-control repository, for which the master repository is presently hosted on the GitHub website,

https://github.com/landlab/landlab. Release versions are also freely available through the *pip* and *conda* Python package managers. The model repository maintained by CSDMS offers links to Landlab documentation and to the GitHub repository, increasing Landlab's visibility to the surface process modelling community in particular. Web-based documentation is hosted at http://landlab.github.io. This includes both developer-written summary documents and tutorials, as well as reference-level documentation that is automatically generated from inline comments and examples in the code itself.

**Programming language.** Landlab is written in Python and exploits and includes as dependencies a number of widely used scientific Python packages: numpy, scipy, matplotlib, nose, netCDF4, numpydoc, cython, six, pyyaml, setuptools, and libgcc. The decision to write in Python was explicitly made to lower the bar for entry to Landlab, to increase the flexibility and reusability of the code base, and to increase development speed both for the core development team and for future users.

Informal canvassing amongst the surface process community, especially amongst graduate students and other early-career scientists less likely to already be strongly wedded to a certain development environment, revealed a marked preference for – and greater familiarity with – Python over C++ (other open-source languages were rarely mentioned). This changing preference for Python has also been noted for PhD students in general, beyond just the field of surface process modelling (Chue Hong, 2014). The choice of Python also means that developers using Landlab can take advantage of that language's

affinity for rapid development (Prechelt, 2000). In particular, Python's dynamic typing and interpreted rather than compiled implementation remove the developer's need to deal explicitly with memory management (van Rossum and Drake, 2001). Other advantages of this choice include high portability between platforms, open-source language, numerous existing scientific libraries, and support for selective optimization of time-critical parts of the code base using Cython and/or compiled-language extensions. Cython is a compiled language that is a super-set of Python, and Cython extension modules

interact seamlessly with pure Python. However, program modules written in Cython allow more granular control of memory management than is the case in pure Python, which can result in significant acceleration of code. Cython is already in use within Landlab for sections of the code that require long out-of-sequence iterations through arrays, and other sections where pure Python would tend to have poor performance. For example, Cython is used in the construction of some of the grid element connectivity arrays, in the FlowRouter and FastscapeEroder components, and in the CellLab extension to Landlab

(Tucker et al., 2016).

**Code sustainability.** A key objective for Landlab from inception has been that the code base be sustainable (Adams et al., 2014; Becker et al., 2015; Katz et al., 2015; Stewart et al., 2010). Following other authors, we view sustainable software as that which is able to continue effectively, sustaining or improving its functionality through time while at the same time



adding new users. Stewart et al. (2010) drew attention to a number of key features of sustainable software, which we have sought to implement. These are:

- *Strong, consistent leadership.* The authors of this paper represent the core development team of Landlab.

- *Rapid prototyping and evolutionary design.* Landlab was initially developed to fill the immediate research needs of the core development team, giving it a strong and well-defined initial direction. In this initial development phase, we have emphasized long-term mountain belt evolution modelling; steady- and nonsteady-flow routing; eco-, surface, and shallow subsurface hydrology; hillslope dynamics; cellular automaton modelling; vegetation dynamics; and ecosystem dynamics. However, the explicitly modular nature of Landlab means that it can readily adapt to new scientific objectives and expand to meet new and as yet unforeseen demands in the future.

- *Modern and effective software engineering practices.* Landlab takes advantage of a number of best practice processes, including extensive and automated unit testing of key code functionality, a formal bug- and issue-tracking record implemented through GitHub, cross-team review of code changes before they are merged into the master branch, and thorough code documentation. A significant portion of our online documentation is created semi-automatically from inline code comments. This reduces duplication of information and aids maintenance and updating of the documentation as the code changes. Individual functions and classes are documented automatically using Python's docstring functionality. General descriptive documentation and tutorials are created and maintained manually. Auto-generated documentation is updated and posted to the project website automatically as new code changes are committed to the GitHub repository using "webhook" functionality provided through the http://readthedocs.org website.

- *Sustained compatibility with underlying libraries, protocols and operating systems.* Landlab is compatible both with Python 2 and 3. The code base is tested automatically using Travis (Mac, Unix) and Appveyor (PC) continuous integration platforms, across Python versions 2.7, 3.4, and 3.5 (see also Section 4).

- *Dissemination and community understanding.* We have sought to publicize Landlab widely at a number of international conferences and workshops, classes, and through collaborative networks. We estimate that at as of mid-2016 approximately 330 potential users have now seen or participated in Landlab-based presentations or classes.

- *Encouraging collaborative software development.* Landlab enables users to tailor its functionality to their specific needs, through its modular design and flexible grid and grid functions. We are already aware of a number of groups outside the core Landlab development team working with Landlab for their own research purposes.

A secondary aspect to sustainability is the ability to have the software continue to be useable after the active development cycle has ceased (Stewart et al., 2010). We anticipate that the choice of Python, minimal system and extension





package requirements, open-source availability of our code base, and thorough documentation will sustain our code for the foreseeable future.

## 3 Model Architecture

Landlab has an essentially tripartite structure – a core grid module, a library of process components, and a set of supporting utilities (Fig. 2). The various subdivisions of the code behave as Python modules, and can be imported and used within a Python environment independently.

### 3.1 Landlab's Gridding Engine

Landlab provides the ability to create a two-dimensional simulation grid of a user-specified size and shape, with a single line of code. Grids are represented as Python objects; a grid object includes data describing its geometry and topology, as well as a variety of methods and functions to manage data and perform common numerical operations. (In object-oriented programming parlance, a *method* is a procedure associated with an object; in this case, "method" means a function that is defined within the grid class, and that can be accessed with the "`grid.method()`" syntax typical of other class
properties.)

### 3.1.1 Grid types and elements

A Landlab grid is defined by a set of grid primitive elements: nodes, links, cells, corners, faces, and patches (fig. 3). In terms of graph theory, these can be thought of as two interlocking and offset sets of points (nodes vs. corners), edges
(links vs. faces), and areas (patches vs. cells). The entire grid can be generated from a description of the geometry of only one of these element types – typically, a user might specify the locations of the nodes, and the grid object's remaining elements are automatically placed according to this node framework.

Each element type shares unique one-to-one or one-to-many geometric mappings with the other elements. Were the
grids to be infinite, these mappings would be perfectly reciprocal – the topology and connectivity of each element with respect to every other element would be identical everywhere it occurs. However, because these grids are finite, we must arbitrarily decide whether the bounding elements are nodes-links-patches or corners-faces-cells. We have chosen the former (see Figs. 4, 5), which means that for example, while all cells have nodes, not all nodes have cells – as the nodes at the grid perimeter cannot have cells defined around them. Table 1a lists the unique one-to-one mappings of features, and emphasizes





which element defines the grid edge in each case. Table 1b lists the primary one-to-many relationships defined for each element type, and lists the standard number of mapped elements (if well defined) for each of the primary grid classes. Note that this table only lists the most useful identities within the three-element groupings node-link-patch and cell-face-corner. The other identities also exist, and can be reconstructed from the one-to-one identities in Table 1a.

Data can be assigned to any element of the grid (see Section 3.2, below). The grid classes also provide properties that define and describe the geometric interrelationships amongst these grid elements (see, e.g., Fig. 4). These mappings allow common geometric operations (such as calculation of gradients across the grid, finding maximum/minimum/mean values of neighbors, upwinding schemes, and flux divergences) to be achieved in typically one or two lines of code.

Landlab provides native support for both regular and irregular grids (Figs. 3, 4). Treating both grid types natively within Landlab allows the grid to be tailored to specific applications. For example, raster grids provide compatibility with digital elevation model data, and can in some cases allow better optimized process algorithms. Trigonal grids with hexagonal cells provide an additional axis of symmetry, and obviate the need for handling diagonal connections in certain types of numerical algorithm (such as flow routing, e.g., Jenson and Domingue, 1988). Irregular grids avoid some of the cardinal direction artifacts than can form on regular grids, such as linear networks and linear drainage divides (Braun and Sambridge, 1997).

Regular grids with quadrilateral cells are implemented as rasters, and irregular grids and all other regular configurations (e.g., hexagons) are implemented as Voronoi-Delaunay interlocked meshes, as also used in the landscape evolution models CASCADE (Braun and Sambridge, 1997) and CHILD (Tucker et al., 2001b). Grid subtypes are defined within these broad families (Table 2). Landlab also implements a base grid class ("ModelGrid") from which both the raster and Voronoi-Delaunay grids are derived. This class describes the elements of the grid and allows their geometries and topologies to be set, but defines no rules for how to do this. This base grid class is primarily intended as a framework from which to derive new grid architectures, rather than as a usable grid type in isolation.

Although the grid primitive element set is shared between the various grid types, the implementation of the geometries is slightly different. For example, core nodes in a raster grid will always have exactly four links, whereas they may have any number of links in a Voronoi-centered irregular grid (Table 1b, Fig. 3). Similarly, methods defined for the grid may be polymorphic or overloaded to optimize functionality for each grid type.



### 3.1.2 Grid standardization and conventions

All Landlab grids share an identical scheme for the numbering of their elements. All elements are numbered from the bottom left of the grid, starting with an ID of 0. All features are ordered first by y coordinate, then by x, taking the midpoint (for linear features such as faces or links) or geometric center (for areas such as cells or patches) for non-point elements as necessary (Fig. 4).

For rotational ordering, Landlab adopts the mathematical standard convention of *counterclockwise from the positive x axis*. This applies not only to almost all measured angles (unless otherwise explicitly noted), but also to the ordering of elements around other elements (such as links around a node), and to the ordering of grid edges where needed (i.e., the standard order is right-top-left-bottom edges). Simple ordering examples are illustrated in Fig. 4.

We extend this same rotational convention to define the directionality of all linear elements (such as links and, where necessary, faces), when such directionality is required. The positive direction is associated with the top-right (first) quadrant; in other words, the positive direction is the one that points more right than down or more up than left. This is shown in more detail in Fig. 4b. This kind of directionality is important for example in the definition of fluxes along links into and out of nodes. In the case of link directions, Landlab provides masking arrays that can describe the local orientation of each link with respect to another feature; for instance, `link_dirs_at_node` describes whether a link points into (+1) or out of (-1) any given node. The use and utility of such data structures is illustrated in Section 5.

### 3.1.3 Mappings and grid characteristics

Landlab uses a standardized grammar to describe the methods and Python properties in the grid classes that provide information about the mapping of grid elements onto other elements, and to obtain information about the grid (e.g., areas, lengths, gradients). The intention of this standardization is to both make it easier for users to quickly find the method they require, and also to provide information on the computational efficiency of the operation. Some of this information is summarized in Table 3.

***Grid characteristics.*** Landlab grids provide Python properties to describe the geometric characteristics of the elements themselves, for instance, position, dimension. These properties are denoted by the preposition "of", as in, for example, `width_of_face`, `length_of_link`, and `area_of_cell`. Use of the word *of* tells the user that an array of floats (or, more rarely, integers) denoting a grid characteristic is the expected return. (See for example use of `angle_of_link` in Fig. 4b.) *Of* is also used to access many counted characteristics of the grid as a whole, such as `number_of_nodes`. All these properties return pre-allocated arrays or single values already stored in memory, and can be expected to be fast.





***Grid element mappings.*** The grid also provides numerous Python properties that describe the connectivity and associations of elements with one another. These are denoted by the preposition *at*. Examples include `face_at_link`, `link_at_face`, `links_at_node`, `patches_at_node`, and `node_at_cell`. Use of *at* tells the user that an array

of element IDs is the expected return (see Fig. 4 for examples of usage). The Landlab boundary condition interface also uses *at*; for instance, `status_at_node` returns an array containing the boundary-condition status (as an integer code) of the grid nodes. All these properties return pre-allocated arrays, and can be expected to be fast.

***"has", "is", and "are" methods.*** Use of *has*, *is*, or *are* in a method name indicates that the method in question applies a

logical test to grid properties. These are not simple lookups, as in the case of *at* and *of* properties, but can still be expected to be fairly fast. The returned object will either be a Boolean, or an array of Booleans. Examples include `is_boundary`, `are_all_core`, and `has_field`.

***"get" and "create" methods.*** Landlab's design philosophy seeks a balance between speed of access of information about the

grid, and memory usage. To this end, only the most commonly used arrays of grid characteristics accessed by *at* and *of* properties are created at grid instantiation. In other cases, these arrays are allocated in memory at the first time of usage in code, then referenced from that point on at subsequent calls of the property. Methods in the grid that begin with *get* or *create* are called by these properties the first time they themselves are used, and construct the necessary arrays in memory. These methods are typically intended for call only by a well-defined subset of other methods internal to grid, and not directly by the

user; i.e., in programmer's parlance they are "private". We use the standard Python practice of beginning such methods with a leading underscore in the name, which tells the various Python user interfaces not to report them in standard lists of grid methods.

***Analytical methods.*** Landlab provides a large number of grid methods to allow easy completion of common and frequently

repeated analyses of the values on the grid. These are denoted by names that begin with *calc*, to denote methods that calculate a new value from provided data, or *map*, which apply some standard rule to map multiple values for connected elements to a single value on the shared element to which they connect. For instance, *calc* methods might allow calculation of gradients at links from data defined at nodes (`calc_grad_at_link`), or flux balances at a node from fluxes defined at incoming and outgoing links (`calc_flux_div_at_node`). *Map* methods might return means of values at links around

nodes (`map_mean_of_links_to_node`), or minima of node values attached to each link (`map_min_of_link_nodes_to_link`), or the maximum slope of links leaving each node (`map_downwind_node_link_max_to_node`). More complex mapping schemes are also available, to allow for instance the mapping of data from topographically upwind or downwind elements only (for example,



`map_value_at_upwind_node_link_max_to_node`). All these methods require active calculation and memory allocation of new values.

***Boundary condition control.*** Grid methods that allow user control of boundary conditions use the word "*set*". Boundary
condition handling is described further in Section 3.1.4, below.

***General rules.*** Words are separated by single underscores. Nouns are typically singular, both describing the element and its characteristic, e.g., `area_of_cell`, not `areas_of_cells`. The exceptions are cases in which more than one thing is associated with each element, such as `links_at_node`, `faces_at_cell`. Any grid property can be expected to be a
fast lookup operation if called repeatedly; methods may require additional memory allocation.

### 3.1.4 Grid boundary condition handling

Also provided are methods to facilitate boundary condition handling (Fig. 5). Nodes can have one of four boundary condition types: *fixed value* (Dirichlet), *fixed gradient* (Neumann), *looped*, or *closed*. A node that is not defined as a boundary is known as a *core* node. The boundary conditions defined on the nodes determine whether each connecting link is
*active* (allows flux along it), *fixed* (allows flux, but flux value is fixed) or *inactive* (flux is forbidden), as shown in Table 4a. Each of these boundary conditions is associated with an integer value, which can be seen in the boundary condition arrays `grid.status_at_node` and `grid.status_at_link` (Table 4b).

We should emphasise that this framework is provided for user's convenience; it can be easily ignored if a user
wishes to implement a different scheme for boundary condition handling. Further, the appropriate boundary conditions depend on the physical scenario that the user is modeling.

The edges of a Landlab grid are always defined by boundary nodes. Because perimeter nodes lack cells (Section 3.1.1), this means not every boundary node necessarily has a cell, and may also not have the standard number of links,
patches, etc. (Table 1b). Conversely, any core node can always be expected to have a cell and a standard connectivity as described in that table. Likewise, inactive links at the grid perimeter lack faces, but each active link always intersects, and is uniquely associated with, a single face (Fig. 5). Thus cells share the boundary conditions of nodes (core vs. boundary) and faces share the boundary conditions of links (active vs. inactive). Note also that nodes that are in the interior of a grid (i.e. not perimeter nodes) can also be assigned as boundary nodes, and that whether or not this occurs depends on the shape of the
area that the user is modeling. For example, a user may wish use a grid that represents a drainage basin, with the basin's interior consisting of core nodes, a single node representing the outlet (flagged as a fixed-value or fixed-gradient boundary), and the remainder of the nodes flagged as closed boundaries.





The grid itself is responsible for keeping track of and ensuring internal consistency between boundary condition properties. The standard numpy setters and getters are overridden for the boundary condition data structures to ensure this internal consistency without the user's involvement. For example, if a user changes a node's status from core to fixed-value

boundary, the gridding engine will automatically update the status of the relevant links.

### 3.2 Spatially distributed data and data fields

A key element of any model of surface processes is a description of how the state variables and surface characteristics vary across the domain. Such data can include both scalar measurements at a point or over an area (such as

topographic elevation, water depth, sediment cover fraction, vegetation type), and directional vector data, for instance, describing fluxes across the surface or gradients in scalar values. Landlab uses data constructs called *data fields* within the grid to store and handle this information.

A prominent advantage of the field system is that data may be associated with any of the grid elements: node, cell,

link, face, patch, or corner. Data fields are one-dimensional numpy arrays whose length matches the number of elements in question. By indexing the array with the ID of desired elements, the values at specific locations and on each element type can be recovered. This scheme readily allows the storage of both scalar and vector data by exploiting the geometric relationships between the node-link-patch (and cell-face-corner, if desired) groupings, as in a traditional staggered-grid scheme (Harlow and Welch, 1965; Slingerland et al., 1994). Scalar data can be stored at nodes. Because links describe the

connectivity between nodes, vector information describing fluxes or gradients between nodes is readily stored on links; the link's orientation provides an implied unit vector, while the associated value represents the vector's magnitude. There are also a number of use cases in which values can usefully be stored on patches, for instance, in representing resolved means of vector values at the bounding links. This data structure also lends itself to the implementation of some cellular automata. For instance, pairwise transition automata (Narteau et al., 2001; 2009) are implemented in Landlab by mapping the pair states

onto the links of a Landlab grid, and representing the corresponding automaton cell states at grid nodes (Tucker et al., 2016).

In terms of implementation in the code, Landlab fields are represented as a dictionary of Python dictionaries within the grid object. The keys to the first dictionary are strings of the names of the grid elements (*viz.*, 'node', 'link', 'patch',

'cell', 'face', 'corner'); the keys to the dictionaries that these return are Landlab *field names*. Users are free to create field names as they wish. However, Landlab maintains a standard format and name list which is widely used by the Landlab component library (supplemental Table S1), and users are strongly encouraged to adopt this scheme to enhance




standardization and interoperability throughout the software. Our standard naming scheme echoes that of the community standards adopted by the Community Surface Dynamics Modeling System (CSDMS). Our rationale follows theirs, aiming to remove ambiguity in the identification of different types of numerical information (Peckham, 2014; Peckham et al., 2013).  However, given the potential for high frequency of name usage in Landlab code, and our ability to easily assess potential

ambiguities between different components, we place more value on name brevity at the expense of total unambiguity as compared with the formal CSDMS Standard Names (https://csdms.colorado.edu/wiki/CSN_Searchable_List). Nonetheless, we maintain one-to-one mappings with the CSDMS Standard Names to enable automated implementation of the CSDMS Basic Model Interface (BMI; see Section 3.4.1).

The general format for Landlab names is "thing_described__quantity_described". This approach is more generally known as the object-attribute-value paradigm: the first word or phrase describes the object, the second word or phrase describes the attribute, and the variable's content is its value. A double underscore separates the object from the attribute. An example might be "surface_water__discharge". A full list of names used in Landlab components as of version 1.0 can be found in the supplementary material as Table S1. A version of this list up to date with the current release version can be

found on the Landlab website.

Units can be attached to grid fields. They are recorded in a further dictionary-like structure, which is a property of the element container. This means they can be accessed with syntax like `grid['node'].units['field__name']`.

Landlab offers some degree of syntactic sugar for its field name interface, as well as convenient shortcuts to create new fields of ones (`grid.add_ones`), zeros (`grid.add_zeros`), and from existing data (`grid.add_field`). For instance, `grid['node']['my_field_name']` is equivalent to `grid.at_node['my_field_name']`.

### 3.3 Components

Components are Python objects that simulate processes within Landlab. A typical Landlab component provides a numerical representation of a single process. For instance, a component might compute the flow of water across a terrain surface using a particular flow law and numerical solution method. Components also exist in Landlab that produce only spatially invariant time series, or that produce time-invariant steady state solutions across a surface. A prominent example would be the FlowRouter component, which calculates the steady state accumulation of water discharge and upstream total

drainage area through a drainage basin. The latter category also includes a number of analytical tools that produce spatial statistics for a surface; for example, components to calculate the steepness (Wobus et al., 2006) or Chi index (Perron and Royden, 2012) for a channel network.




Multiple components can be used together, allowing the simulation of multiple processes acting on a single grid. For example, components simulating hillslope processes and fluvial geomorphic processes can be easily implemented together to create a "custom" landscape evolution model. In some cases, the output from one component may form the input

to another, as for example when combining flow routing and sediment transport components, or soil moisture and vegetation growth components. The design of each component is intended to work in a "plug-and-play" fashion, where each component couples simply and quickly to others. This is permitted by a standardized interface for each component, as described in Section 3.3.1. Examples of coupled component systems can be seen in Section 5.

Landlab provides a suite of existing components that can be deployed by users. Future versions of Landlab will add further components designed by the core development team. However, we anticipate that users of Landlab will also devise new components of their own, allowing the exploration of new processes within Landlab. In keeping with the open source ethos of the project, we would encourage such users to in turn commit their work back to the master fork of Landlab, for the use of others. Documentation and advice for this process can be found on the Landlab website.

### 3.3.1 Component standard interface

Landlab components have standardized interfaces, which are designed to enhance interoperability both internally to Landlab (between components, or between components and Landlab utilities) and between Landlab and external interfaces like the CSDMS Basic Model Interface (Peckham et al., 2013) (see also Section 3.4.1). The Landlab standardized

component interface consists of the following:

- An initialization method, with the standard argument signature `__init__(self, grid, x=a, y=b, z=c, ..., **kwds)`, where *grid* is a Landlab grid object, *x*, *y*, and *z* are component-specific keyword arguments with default values *a*, *b*, and *c*, and ***kwds** is an optional keyword argument dictionary. The grid object passed during instantiation is accessed during the running of the component, and its data fields are updated automatically. A

component may have any number of component-specific keyword arguments. The variable names of these arguments are not standardized, but rather are generally unique to each component. The component-specific arguments are, however, required to have default values. The names of the keyword arguments make explicit the data requirements of the component in order to run. However, the ***kwds** argument alternatively allows these parameters to be set from a dictionary of model parameters. In other words, this component could be initialized in

two equivalent ways:

```
>>> ld = LinearDiffuser(grid, linear_diffusivity=1.0, method='simple')
```
or





```
>>> paramdict = {'linear_diffusivity': 1.0, 'method': 'simple'}
>>> ld = LinearDiffusivity(grid, **paramdict)
```

- A run method, with the standard argument signature `run_one_step(dt, *args, **kwds)`, where *dt* is an
  interval of time over which to execute the component before returning a result, and *\*args* and *\*\*kwds* are an
  argument list and dictionary respectively, specific to each component. These latter items allow any additional
  arguments necessary for the model to run to be passed in. If *dt* is not required for a component to run, it may be
  omitted.

- A standard set of properties for the component: *name*, *input_var_names*, *output_var_names*, *var_units*,
  *var_mapping*, and *var_definition*. These properties describe the fields that the component interacts with, the units of
  each, which element each field is defined on, and a brief summary of what each field represents.

All components inherit from the base class *Component*. This base class enables and regulates the standardized
properties and interface that are available for every Landlab component. It also provides methods designed to streamline the
creation of the output data fields when a component is instantiated.

Landlab version 1.0 provides a standard component library as part of its installation. A full list of components
available in version 1.0 can be found in Table 5. However, Landlab permits modeling of the evolution of almost any two
dimensional system that lends itself to description by discretized systems of differential equations or cellular automaton
rules.

### 3.3.2 Timestepping and interaction of components

For most existing Landlab components, the component is responsible for controlling its own internal numerical
stability. A timestep parameter *dt* is passed to each component that operates in a time-dependent fashion; this timestep can be
thought of as the "coupling timescale", and represents the frequency of interaction between components if more than one is
coupled (Fig. 6). However, it is not necessarily the stable timescale, which will vary between components. Each component
is responsible for calculating its own stable timestep under the model run conditions, and internally subdividing the imposed
*dt* in order to ensure the model run does not become unstable. The user is responsible for selecting an appropriate coupling
timescale – too short, and a model run will take more steps than necessary for each component to be stable; too large, and
information transfer between the components will be limited, possibly introducing an additional source of numerical error.



This serial coupling approach has the advantage of simplicity, but has the disadvantage that each component is treated independently for the duration of a time step, rather than fully coupled. Future development may allow tighter time coupling between components. Rather than passing a fixed *dt* to each component in turn, we envision that components could optionally return Jacobian matrices, which could be combined externally and inverted together using a coupling script. This

may be an option in future versions of Landlab.

### 3.4 Utilities and interfaces

In addition to the grid, which governs the topology and connectivity of spatial data, and the components, which describe how spatial data changes with time, Landlab also offers tools that control input and output, including data input and

export, translation between widely used data formats, plotting, and the BMI external model interface. Landlab can read and write data files in NetCDF4, VTK, and ESRI ASCII data formats. These options are intended to allow interoperability with third-party software, especially Geographical Information Systems, and also to allow Landlab data to be manipulated in and displayed with specialized visualization software (such as ParaView).

Landlab's standard interfaces also allow it to interact more easily with software frameworks developed by the geoscience and hydroscience communities. For instance, Landlab is already embedded within the Hydroshare collaboration environment, http://www.hydroshare.org. This means that Landlab models can be created and run within the Hydroshare data and modelling environment, and can take advantage of that environment's shared data platform and metadata systems.

### 3.4.1 Dynamic model interaction and the Basic Model Interface

As noted in previous sections, Landlab has been designed from conception to be fully compliant with the Community Surface Dynamics Modelling System's Basic Model Interface (BMI) (Peckham et al., 2013). The BMI concept allows any two models describing the changes caused by surface processes to be coupled together, regardless of the vagaries of model gridding schemes, programming languages, or other low-level design choices. It does this by means of a standard

interface (the Basic Model Interface, *sensu stricto*), which is callable for any BMI compliant model or component and includes generically applicable functions such as `initialize`, `update` (i.e., run one timestep), and `get_current_time`. The interface allows information about the current state of a simulation to be passed back and forth between running models in a manner that is agnostic in terms of implementation details.

The Landlab framework is designed such that the Landlab standard component interface can also expose a full BMI interface; in other words, *all Landlab components are also BMI-compliant components*. This means that by choosing





Landlab as their model development environment, users also gain the ability to couple their models immediately with any other model in the CSDMS repository of BMI-compliant codes. This choice will also enhance the utility of Landlab to users who wish to implement component functionality alongside some other model using the CSDMS BMI or Web Modeling Tool (WMT) (Piper et al., 2015).

**4. Validation, testing, and documentation**

Landlab makes extensive use of Python's native documentation and code testing systems in order to test and validate the code base and to keep our documentation up to date. The development team exploits a combination of this Python "doctesting" and unit testing techniques to simultaneously test and document the code base. Doctests are code examples that can be included in the docstring that describes each Python method, and they list the expected output from

each line of code as part of the documentation. Crucially, this code is then actually run whenever testing of the code base is triggered (for instance, by calling `landlab.test()`), and any doctests for which the output does not match the expected solution are recorded as either an error (tested function does not run cleanly) or a fail (output does not match). Because doctests are part of each function's docstring, they are also then automatically scraped from the code and included in the online documentation as examples for the user. In this way doctests allow us both to help ensure Landlab functionality does

not break as the code base evolves, while at the same time documenting for the user the way in which a given method, function, property, or component can be used.

Landlab also includes suites of unit tests. These are test scripts written specifically to exercise particular aspects of the code, and to check the output of that test against known correct solutions. Examples of when this is useful can occur in

longer or more involved code, especially in components, where various different configurations of grid types and initial and boundary conditions need to be tested to ensure the component is robust under various different conditions. Unit tests differ from doctests in that they are not intended to be user-facing, although they are run alongside them when testing of the code base is triggered.

Almost all core Landlab functionality of both grid methods and components is now tested in this way. As of this version, around 1400 separate tests are run on the code each time testing is triggered, and the tests cover 80% of the code base. Most of the remaining uncovered code is either challenging to adequately test (for example, plotting functions), not part of the core Landlab functionality (such as helper scripts involved in building releases), or deprecated. Tests are triggered automatically and remotely through the web-based applications Travis (Mac/Linux) and Appveyor (PC) whenever a new

commit is made to either a branch or the master version of the code repository on GitHub, or when a new release of the code is built. These tests are performed on a range of supported Python versions, including both versions 2 and 3. Tests can also




be triggered manually on a local machine by running a testing script included with Landlab, or by calling `landlab.test()` from an interactive Python environment.

## 5. Creating models with Landlab

We here illustrate some of the key functionality of Landlab by example, demonstrating its applicability across a variety of types of problem. We hope to emphasize here that Landlab *is not a landscape evolution model* (although it can be used to create them) – rather, it presents a framework under which a wide variety of different models can be implemented using its tools, including hydrologic, ecologic, and sedimentological models, as well as landscape evolution models. This section illustrates four possible contrasting model designs that can be implemented within the Landlab framework: a very

simple "toy" geomorphic diffusion code that demonstrates the core functionality of the grid; a coupled stream power-hillslope diffusion model driven with a stochastic sequence of storms, illustrating some of Landlab's components; a cellular automaton, demonstrating a fundamentally different style of model implementation that is also enabled by Landlab's design; and a flood wave routing model, run on real topographic data ingested by Landlab.

**5.1 A simple diffusion model**

Although Landlab provides "off the shelf" process simulation code in the form of the components, Landlab also facilitates the design of models without using the components. The Landlab grids provide mapping, gradient, and divergence functions to make implementation of, for instance, finite-difference or finite-volume methods both concise and straightforward.

Here we illustrate this functionality using a simple finite-volume diffusion scheme, which here is representing the down-slope flow of soil on hillslopes (Culling, 1963). We wish to represent the evolving form of a diffusional hillslope that is undergoing a constant uplift (1 mm y$^{-1}$) with reference to a relative base level. In this case, the grid is radial and so roughly circular in plan view. Use of this particular configuration is intended in part to demonstrate the flexibility of Landlab's

design, although this radial grid arrangement could perhaps be thought of in terms of response to a rising volcanic mound or salt diapir, or other similar radially symmetric uplifting scenario.

The governing equations for this example are:

$$\frac{\partial \eta}{\partial t} = U - \nabla q_s$$



(1)

$$q_s = -D\nabla\eta$$

(2)

where $\eta$ is land-surface elevation, $t$ is time, $U$ is the rate of vertical motion ("uplift") of rock relative to baselevel, $q_s$ is volumetric sediment flux per unit slope width, and $D$ is a transport coefficient with dimensions of length squared per time.

For our example model, Eq. (2) will be discretized and solved using a finite-volume solution scheme. Consider a cell of surface area $a$ that is surrounded by $N$ neighboring cells (Fig. 7). We can integrate Eq. (1) over the surface area of the cell:

$$\int_a \frac{\partial\eta}{\partial t} da = \int_a U da - \int_a \nabla q_s da$$

(3)

Applying the divergence theorem to the last term on the right, and evaluating the other two integrals,

$$a\frac{\partial\bar{\eta}}{\partial t} = Ua - \oint_p q_s(p)\,\mathbf{n}\,dp$$

(4)

where $\bar{\eta}$ is the average elevation within the cell, $p$ represents position along the perimeter of the cell, and $\mathbf{n}$ is a unit vector

normal to the perimeter and pointing outward. The last term is a line integral that represents adding up all the inflows and outflows of mass along the cell's perimeter. If the cell is a polygon with $N$ faces, this last term can be replaced by a summation:

$$\frac{\partial\bar{\eta}}{\partial t} = U - \frac{1}{a}\sum_{k=1}^{N} q_{sk}w_k$$

(5)

where $q_{sk}$ is the unit flux at face $k$, positive outward, and $w_k$ is the width of face $k$.

We will implement this solution in Landlab by assigning to each node $i$ the value of the average elevation within its cell, $\bar{\eta}_i$ (for notational convenience, we will drop the use of the overbar below). To calculate the flux at each face, we first need to calculate the topographic gradient at each face. We will do this by taking the elevation difference between each

neighboring pair of nodes, dividing by the length of the link that connects them, and then assigning the resulting gradient value to the relevant link. The gradient at link $j$ is therefore calculated as:




$$G_j = \frac{\eta_{H_j} - \eta_{T_j}}{L_j}$$

(6)

where $\eta_{H_j}$ and $\eta_{T_j}$ are the elevation values at link $j$'s head and tail nodes, respectively, and $L_j$ is the length of link $j$. In Landlab's gridding engine, the calculation of link-based gradients in a node-based scalar quantity like $\eta$ is handled by the grid method `calc_grad_at_link`, which takes a node array or field name as an argument and returns a link array.

Figure 7 illustrates how values of $\eta$ defined at nodes can be used to calculate gradients at links, and then the gradients can be used to calculate the net flux into and out of a cell.

In our diffusion example, the summation of fluxes along the cell faces is calculated as follows:

$$\sum_{k=1}^{N} q_{sk} w_k = \frac{D}{a_i} \sum_{k=1}^{N} \delta_{ik} G_k w_k$$

10    (7)

where $\delta_{ik}$ indicates the direction of link $k$ relative to the cell $i$: if $\delta_{ik}$ = -1, the link points outward from the cell; if $\delta_{ik}$ = +1, the link points inward.

To calculate flux divergence using this finite-volume approach, Landlab provides the general grid method
`calc_flux_div_at_node`, which takes a link-based array of unit fluxes as an input, and returns a node array that contains the sum of in/out fluxes (divided by cell area) at each node (Fig. 7). Values at perimeter nodes, which lack cells, are ignored. In keeping with the standard definition of the divergence operation, the function returns positive values where the net flux is outward, and negative values where it is inward.

In the diffusion example shown in Figure 8, the time derivative is discretized using a simple forward-Euler explicit method, such that the values of elevation at the new time step $t$+1 are calculated from values at the old time step:

$$\eta_i^{t+1} = \eta_i^t + \Delta t \left[ U + \frac{D}{a_i} \sum_{k=1}^{N} \delta_{ik} G_k w_k \right]$$

(8)

where the superscript indicates time step, and the quantity in brackets is evaluated at time step $t$. The code to implement the
model is shown in Figure 8. Note the use of the `calc_grad_at_link` and `calc_flux_div_at_node` methods (and note also that $U = 0$ in this example).





An advantage of the finite-volume approach is that it can be applied to cells of any shape. For instance, it can be used with hexagonal cells, or with Voronoi polygons as in the example in Figure 8.

This model can be implemented in Landlab and plotted in as few as 16 lines of code (Fig. 8). Here, line 1 imports

the Landlab classes and functions we will use, and line 2 imports the `show()` function from matplotlib that will let us display the plot. Line 3 instantiates the Landlab grid object. This example uses a RadialModelGrid, but the same code would work with any grid type. Lines 4–6 initialize data for the model run. $z$ will be the land surface elevation at each node; $qs$ will be the volumetric sediment flux per unit width along each link. Note that this implementation is consciously not using data stored as Landlab fields, to illustrate that this is not a requirement; however, it would be trivial to modify lines 4 and 5 to

create the data as fields on the grid, and the remainder of this script would be unchanged. Line 7 is the first line that actually begins the calculations that perform the diffusion. This line calculates a Courant-Friedrichs-Lewy (CFL) stability condition (Slingerland and Kump, 2011) for the maximum stable timestep for the finite-volume scheme we are about to implement.

Lines 8–14 implement a time loop, within which the diffusion occurs. The core (i.e., interior) nodes of the grid are

uplifted at a rate of 0.001 length units per time unit relative to base level. Lines 10–14 implement the meat of the differencing scheme, where we use a staggered grid to solve the discretized diffusion equation (Eq. 8). The fluxes on the links are calculated as the product of the diffusivity parameter $D$ and the topographic gradient at the links (lines 10, 11), taking care to calculate the flux only on active links. The flux divergence is then calculated at each node based on the fluxes on the links to which is it adjoined (line 12). Note that Landlab enables each of these operations to be performed with a

single grid method. The final lines of the code invoke the standard Landlab plotter, then display the output. Although we have not specified any particular units in our calculation, in line 15 we assert that the length unit is meters and the time unit is years.

Note that this same result could have been achieved even more concisely using Landlab's inbuilt LinearDiffuser

component. The equivalent code is shown in Fig. 9. Not only are the implementation details of the scheme now handled entirely within the component, but so also is internal subdivision of the provided timestep to meet the necessary stability conditions for the simulation. Additionally, the elevation data are now passed into the component as the field 'topographic__elevation' – which is attached to the grid – rather than as a separate variable (lines 5, 7), as discussed in Section 3.2.



### 5.2 Coupling diffusion to stream power with a storm sequence

The next example illustrates a simple model for the evolution of an eroding and uplifting landscape, explicitly representing channel incision and hillslope processes. In this model, we also explicitly represent time variability of water input to the system (i.e., storms), based on the conceptual framework of Eagleson (1978) and broadly equivalent to the

approach for modelling storms used in the CHILD landscape evolution model (Tucker and Bras, 2000; Tucker et al., 2001b). In technical terms, the example is designed to show in more detail the use and coupling of several Landlab components: the FlowRouter, the StreamPowerEroder, the DepressionFinderAndRouter, the LinearDiffuser, and the PrecipitationDistribution classes. The aim here is to demonstrate how Landlab couples components, and to illustrate several different component styles.

Here, channel incision processes are represented by the stream power law (Howard, 1994; Lague, 2014; Whipple and Tucker, 1999), which says that incision rate, $E$, of a stream is proportional to a product of powers of channel discharge, $Q$, and local channel bed slope, $S$. In this version, we also include an incision threshold, $C$, below which incision is forbidden:

$$E = K\,Q^m\,S^n - C \qquad\qquad \text{if } C < K\,Q^m\,S^n$$
$$E = 0 \qquad\qquad\qquad\qquad \text{if } C >= K\,Q^m\,S^n$$

$$(9)$$

In this case $m = 0.5$, $K = 1e\text{-}5\ \mathrm{m^{-0.5}y^{-0.5}}$, $C = 1e\text{-}5\ \mathrm{m\ y^{-1}}$, which are fairly typical and widely-adopted values for a generic erosional upland landscape (Harel et al., 2016; Tucker and Whipple, 2002). Here we also adopt $n = 1$. This is primarily to maintain dimensionally sensible units for $K$ while still honouring the widely-observed ratio of $m/n \sim 0.5$. Nonetheless, we note $n > 1$ in some global data compilations for stream power where $C = 0$, and suggest our incorporation of an explicit erosion threshold makes our choice of $n = 1$ reasonable (Harel et al., 2016). We shall see that this set of values

together produce a plausible total landscape relief of order 1 km for catchments of maximum length ~5 km, which is within the range expected for real catchments of this scale in tectonically active regions. Other possible forms of stream power-based incision rules are also possible using this component, but are not illustrated here.

The Landlab StreamPowerEroder and FlowRouter components deployed here use the "Fastscape" algorithms of

Braun and Willett (2013). This solution scheme is implicit and order-n, and permits arbitrarily long, numerically stable timesteps to be taken. The Fastscape algorithm requires out-of-order (i.e., upstream order) iteration through the nodes, but pure Python code has relatively poor speed performance when executing explicit loops or iterations through arrays. For this reason, both the stream power and flow routing components also use compiled Cython (see Section 2.2) to accelerate these





speed bottlenecks in the code. (The release version of Landlab distributes this code in pre-compiled form to users.) The run method of the component performs as order-n, and as expected is unaffected by grid type (in this demonstration, raster versus hex grids). The initialization of the grid and components adds a very small overhead which also increases broadly linearly with grid size (Fig. 10; code in supplementary information as Script S2). This overhead reflects the calculations

necessary to build the data structures describing the grid's connectivity, and is significantly greater for Voronoi grids compared to rasters, due to the iterative calculations required to assemble Voronoi grid-connectivity arrays.

The final topographies from the raster and hexagonal implementations of this pure stream power component are shown in Fig. 11. The code can be seen in the supplemental information as Script S3. It conforms to a typical form for a

Landlab driver script, which looks like:

   1.  Import necessary Python libraries, including from Landlab
   2.  Instantiate a grid object
   3.  Create input fields and set the grid initial and boundary conditions
   4.  Instantiate the components

5.  Perform a loop to run the components
   6.  Finalize, plot, save, and/or export

In this case, the model is driven by a stochastic storm generator (the PrecipitationDistribution class), based on that suggested by Eagleson (1978) and similar to the one underlying the CHILD landscape evolution model (Tucker and Bras, 2000; Tucker et al., 2001b). Unlike CHILD but in keeping with Eagleson's original derivation, here an explicit inverse

relationship between storm length and intensity is built into the distribution, by calculating storm depth as a gamma-distributed random variable, and then deriving storm intensity as the quotient of depth and (exponentially distributed) duration. This approach prevents unrealistic long-duration, high-intensity events from being sampled (Eagleson, 1978). The PrecipitationDistribution class provides a method that yields tuples of interval durations and rainfall intensities as a true Python generator – in other words, fresh values are returned each time the method is looped upon, until such time as the total

elapsed time exceeds the supplied run time, at which point the loop will cease (see lines 46–53 in the code). This makes the implementation of the "run" loop both efficient and concise, as well as being a classically "Pythonic" way to implement this kind of loop. In this instance, the parameters for the PrecipitationDistribution have been chosen to represent a mean annual rainfall rate of around 5 m y$^{-1}$, and where it only rains around 10 % of the time.

The switch between grid types involves changing a single line of code (see the logical test at lines 15–18). Note that although the total number of nodes and the number of rows and columns is identical in both cases, the hexagonal grid is rectangular rather than square due to the single axis of mirror symmetry present in a tessellation of regular hexagons. (The HexModelGrid class provides flags allowing control both of the orientation of this symmetry axis, and also the shape of the perimeter of the grid – rectangular or hexagonal.)



The addition of the linear diffusion component, LinearDiffuser, is performed simply by creating an instance that class, then incorporating its run method into the loop (code S4, lines 40 and 49). As in previous examples, each component is responsible for managing its own internal numerical stability – in this case, if the LinearDiffuser run method receives an
input *dt* that exceeds the Courant-Friedrichs-Lewy stability limit, that timestep will be internally subdivided as necessary within the component.

In this example, because diffusion can occur independently of stream incision, it is possible that diffusion can sever the flow paths of the FlowRouter and create internal basins. Because of this possibility, this version of the code also includes
a lake-filling algorithm, implemented as the component DepressionFinderAndRouter. The lake-filling algorithm identifies closed depressions in the topography then reroutes flow across them, and is based on the algorithm of Tucker et al. (2001b). The final topography of the coupled stream power and linear diffusion model is shown in Fig. 12.

### 5.3 Landlab as a cellular automaton

Much of this manuscript focuses on Landlab as a tool for the implementation of numerical solutions to two-dimensional partial differential equations, as many geomorphic process laws (sensu Dietrich et al., 2003) have been couched in the language of differential equations. However, Landlab can also act as a powerful environment for the implementation of cellular models. Landlab provides a set of tools for the construction of "continuous-time stochastic" (CTS) cellular automata (CA). This interface within the main body of Landlab is known as CellLab-CTS (Tucker et al.,
2016). It enables efficient creation of CTS models: a user needs only to specify the states and transition rules, and write a short Python script to initialize and run a CellLabCTSModel object. Figure 13 shows output from a CellLab-CTS model implementing a lattice-grain algorithm (Tucker et al., 2016).

Landlab can also be used to construct traditional discrete-timestep cellular automata. An example is provided by
developing an ecohydrology model in Landlab (Fig. 14a, code S5), which is in part an implementation of the Cellular Automata Tree-Grass-Shrub Simulator (CATGraSS) (Caracciolo et al., 2016a; 2016b; 2014; Zhou et al., 2013). CATGraSS couples local vegetation dynamics, which simulate biomass production based on local soil moisture and potential evapotranspiration, and plant establishment and mortality based on competition for resources and space at each cell of a gridded model domain. Each cell in the domain can be occupied by one Plant Functional Type (PFT): each cell is flagged as
Tree, Shrub, Grass or Bare (left unoccupied).




CATGraSS is driven by rainfall pulses and solar radiation. In Landlab, the model is implemented as a set of interacting components, each of which describes a different element of the coupled system: PrecipitationDistribution, Radiation, PotentialEvapotranspiration, SoilMoisture, Vegetation, and VegCA. This means that each process can also operate in isolation, outside the context of this example model. The PrecipitationDistribution component simulates the random arrival of storm pulses. Precipitation characteristics are based on the seasonal rainfall statistics of a region, and characterized by using exponential distributions of storm duration, depth, and inter-storm duration. Storm pulses recharge the soil moisture storage, represented as a single bucket (Laio et al., 2001). The Radiation component calculates the ratio of total incoming shortwave radiation on each cell in the domain with respect to a flat surface, assuming homogenous cloud cover, based on local slope and aspect (Bras, 1990). This ratio is used to spatially distribute potential evapotranspiration calculated by the PotentialEvapotranspiration component (Snyder, 2005; Zhou et al., 2013). The SoilMoisture component simulates local inter-storm water balance and root-zone soil moisture saturation depending on the PFT that occupies the corresponding cell at a given time (Laio et al., 2001). The Vegetation component simulates temporal dynamics of above-ground live and dead biomass, as well as Leaf Area Index (LAI). It does this by computing Net Primary Productivity (NPP) based on the concept of water-use efficiency (WUE) that relates NPP to actual evapotranspiration (ET) and vegetation foliage loss due to water stress and senescence (Istanbulluoglu et al., 2012; Zhou et al., 2013). The VegCA component handles the spatial organization of PFTs, through plant establishment, competition, and mortality, by combining deterministic and probabilistic rules. Plant establishment is driven by seed dispersal and water stress, while mortality is related to water stress, plant age, and disturbances (Zhou et al., 2013).

This example ecohydrology model and its constituent components can work both on grids imported from a Digital Elevation Model (DEM) using the `read_esri_ascii` utility (see also Section 5.4), and also on synthetic grids created using the RasterModelGrid library. In the example illustrated in Figs. 14b and 14c, we use the example ecohydrology model (code S5) to simulate plant competition in a semi-arid basin in Sevilleta, New Mexico, USA, modelling the plant species found in this area (Zhou et al., 2013). The domain is initialized with randomly assigned PFTs with random spatial distribution of ages (Fig. 14c(i)). All PFTs initially have an identical cover fraction in the domain. Local vegetation dynamics are simulated at inter-storm time steps, and plant competition is modeled at annual time steps. In the simulations, trees are outcompeted by drought-tolerant shrubs and grasses in the first few hundred years, consistent with regional observations in central New Mexico (Zhou et al., 2013). Shrubs and grasses coexist in the modeled domain with alternating periods of shrub and grass dominance. Note that shrub vegetation shows cluster patterns as they propagate in space by seed dispersal from mature shrub plants.



### 5.4 Landlab as a hydrological modelling environment

Landlab also contains several surface water flow generators, including an explicit two-dimensional solution for the shallow water equations. The OverlandFlow component has been adapted from the flood inundation model described by de Almeida et al. (2012). Their algorithm was derived for use on structured grids, and the Landlab

implementation only works with the RasterModelGrid library. Water discharge is calculated on each active link within the model domain, simulating a hydrograph at each link location.

In many flood-wave routing models, a small time step must be used to prevent instabilities, which often manifest as 'checkerboard' patterns of water depth, from emerging. To maximize computational performance of the

OverlandFlow component, an adaptive time step is used to find the largest time step that adheres to the CFL stability condition (Hunter et al., 2005). To further enhance the stability, the OverlandFlow component also contains stability criteria so that the component can operate not only on shallow, urban areas, but also steeper terrain, such as mountainous watersheds (Adams, 2016). The OverlandFlow component was designed for structured grids, and it assumes water can only move in the four cardinal directions. This is easily accommodated within Landlab, and

several other components (e.g., the FlowRouter, and others in the example presented below) can also be optionally instructed using keywords to only use node neighbors in these cardinal directions.

An example script running the OverlandFlow component can be seen in the supplementary information as Script S6. It follows a similar pattern to scripts outlined in earlier parts of this section, with import of the Landlab and other Python

classes and functions needed, followed by grid creation, component instantiation, component execution in a loop, then finalization and plotting. Notably, this script uses an imported digital elevation model (DEM) of a real landscape over which to route flow, which is ingested into Landlab using the `read_esri_ascii` function contained in Landlab's input and output utilities. Use is made of Landlab's native boundary handling system to designate nodes of the grid outside of the irregularly-shaped catchment as closed, excluding them from the calculations.

This example combines the OverlandFlow component with the SinkFiller. The SinkFiller is run on the initial topography prior to the simulated storm, and fills any local depressions present in the surface. This has been done to enable full drainage of all the water from the network, and to permit evaluation of the full water budget at the outlet. However, in general the OverlandFlow component will happily run on landscapes that do contain pits. In this example, a rainfall rate of

25 mm h$^{-1}$ was run over the watershed DEM for one hour. The resulting hydrograph (water discharge over time) is plotted at the outlet. Water depth across the domain is also plotted to show the wave front propagating downstream (Fig. 15). As expected, the total hydrograph duration is several times the length of the storm, and the peak in the hydrograph lags behind the storm itself significantly, in this case by more than an hour.




### 6. Conclusions

Landlab is an open-source, Python-based software toolkit designed to accelerate the development of new process models. It consists of a gridding engine, a set of components describing individual surface processes, and a set of utilities for data input, output, and visualization. Landlab not only permits the creation of models by combination of existing

components, but is also optimized to aid in the design of new process components. The code base is thoroughly documented both online and within the code itself, and each release undergoes an automated testing procedure to ensure its robustness. A set of tutorials and examples to help learn about Landlab is also provided.

Landlab is explicitly designed to interface with other software, and in particular, with other models of surface

processes. It exposes a CSDMS Basic Model Interface. It can serve as a platform to develop both continuum-based and cellular-automaton-style models, and potentially to have the two model styles interact on the same grid. We illustrate some of the functionality of Landlab and its existing components with a suite of examples drawn from geomorphology, ecology, and hydrology. The examples provided in this paper illustrate the wide diversity of scientific questions that can be addressed using Landlab-built models.

### 15 7. Code availability

This text describes Landlab version 1.0.0, which was released in August 2016. The source code for this version is maintained in a git repository hosted on GitHub at https://github.com/landlab/landlab/releases/tag/v1.0.0 (the latest development version of Landlab is always available at http://github.com/landlab/landlab). Landlab can also be installed as a release version, including pre-compiled binary files containing Cython extensions, through the *conda* and *pip* Python package management

systems, as described in the online documentation. Documentation and installation instructions for the most current release version of Landlab are provided at http://landlab.github.io. Software dependencies are listed at https://landlab.github.io under Install. To the best of our knowledge, Landlab will operate on any system that meets these software requirements; as of the time of writing, Landlab is known to work on, and is tested for, recent-generation Mac, Linux, and Windows platforms running Python 2.7, 3.4 and 3.5. Landlab and its components are distributed under an MIT open-source license.

### 25 Acknowledgements

This research was supported by the US National Science Foundation (ACI-1147454 (GET), ACI-1450409 (GET), ACI-1450338 (NMG), ACI-1147519 (NMG) ACI-1148306 (EI), ACI-1148305 (EI), and EAR-1246761 (through an NCED2 postdoctoral fellowship to DEJH)). Landlab could not exist without the wider open source software in science movement,



and particularly open-source enthusiasts who are members of the surface process modelling community. We are particularly

indebted to the best practices put forward and advice offered by members of the CSDMS Integration Facility.

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





**Table 1a. One-to-one mappings of Landlab grid elements.**

| Element 1 | Element 2 | Behaviour at grid perimeter |
|---|---|---|
| Node | Cell | Perimeter nodes lack cells |
| Link | Face | Perimeter links lack faces |
| Patch | Corner | Neither element defines the perimeter |

**Table 1b. Primary one-to-many mappings of Landlab grid elements.**

| Element | Connected elements | *Number of each connected element in by grid type:* | | |
|---|---|---|---|---|
| | | **Raster** | **Voronoi-Delaunay** | **Hexagonal** |
| Node | Link, patch | 1:4 | Variable | 1:6 |
| Link | Node, patch | 1:2 | 1:2 | 1:2 |
| Patch | Node, link | 1:4 | 1:3 | 1:3 |
| Cell | Face, corner | 1:4 | Variable | 1:6 |
| Face | Cell, corner | 1:2 | 1:2 | 1:2 |
| Corner | Face, cell | 1:4 | 1:3 | 1:3 |



**Table 2. Currently implemented grid types in Landlab**

| Grid type | Grid parent | Notes |
| --- | --- | --- |
| Base | None | The base class; a grid defining the elements but without any internal geometry or topologic connectivity imposed. |
| Raster | Base | Regular grid with identical, square or rectangular cells. |
| Rectilinear | Raster | Regular grid with quasi-rectangular cells whose size can vary across the grid. |
| D8 Raster | Raster | As for raster, but with diagonal links. |
| D8 Rectilinear | Rectilinear | As for rectilinear, but with diagonal links. |
| Voronoi-Delaunay | Base | Irregular grid with polygonal cells and triangular patches. Each node has n>=3 links. |
| Radial | Voronoi-Delaunay | Irregular grid where nodes form concentric, evenly spaced rings around a central node. |
| Hex | Voronoi-Delaunay | Regular grid with identical, regular hexagonal cells and equilateral triangle patches. Each core node has exactly 6 links. |



**Table 3. Standard grid method and property naming conventions, listed in approximate order of operation speed.**

| Name contains | Refers to | Operation speed |
|---|---|---|
| at | Connectivity of grid elements | Lookup |
| of | Property of grid or grid element | Lookup (may require allocation on 1$^{st}$ use) |
| has, is, are | Logical test on grid property | Memory allocation |
| get, create | Memory allocation of grid property | Memory allocation |
| set | Update boundary conditions | Calculation; internal consistency checks |
| map | Map several pieces of data from several elements onto a single element to which they all connect | Several calculations & memory allocations |
| calc | Perform a calculation using data defined on grid elements | Several calculations & memory allocations |





**Table 4a. Link boundary condition status as dictated by node boundary condition status.**

| Nodes at link ends | Link status | Carries flux? |
|---|---|---|
| Core – Core | Active | Yes |
| Core – Fixed value | Active | Yes |
| Core – Fixed gradient | Fixed | Yes |
| Core – Looped | Active | Yes |
| Core – Closed | Inactive | No |
| Boundary-Boundary | Inactive | No |

**Table 4b. Integer values associated with each boundary condition status.**

| Element type | Status | Integer Value |
|---|---|---|
| Node | Core | 0 |
| Node | Fixed value | 1 |
| Node | Fixed gradient | 2 |
| Node | Looped | 3 |
| Node | Closed | 4 |
| Link | Active | 0 |
| Link | Fixed | 2 |
| Link | Inactive | 4 |





**Table 5. Components available in Landlab v.1.0.**

| Component name | Process simulated/Analysis performed | Key reference |
|---|---|---|
| LinearDiffuser | Linear diffusion of topography | Culling (1963) |
| PerronNLDiffuse | Nonlinear hillslope diffusion | Perron (2011) |
| Flexure | Simple lithospheric flexure under loading | Lambeck (1988), Hutton & Syvitski (2008) |
| gFlex | A more complex flexure model, utilizing gFlex | Wickert (2016) |
| FlowRouter | A convergent flow router, following the Fastscape algorithms | Braun & Willett (2013) |
| DepressionFinderAndRouter | A lake filler that can route flow across depressions | Tucker et al. (2001a) |
| SinkFiller | An algorithm to fill depressions in a surface | Tucker et al. (2001b) |
| OverlandFlow | A shallow overland flow approximation | de Almeida et al. (2012), Adams (2016) |
| KinematicWaveRengers | A solution to the depth varying Manning equation for surface flow | Julien et al. (1995), Rengers et al. (2016) |
| SoilInfiltrationGreenAmpt | Infiltrate surface water into a soil following the Green-Ampt method | Julien et al. (1995), Rengers et al. (2016) |
| SoilMoisture | Compute local inter-storm water balance and root-zone soil moisture saturation fraction | Laio et al. (2001) |
| PotentialEvapotraspiration | Calculate potential evapotranspiration across a surface | Snyder (2005), Zhou et al. (2013) |
| Radiation | Calculate total incident shortwave solar radiation | Bras (1990) |
| Vegetation | Calculate above-ground live and dead biomass, and Leaf Area Index | Istanbulluoglu et al. (2012), Zhou et al. (2013) |
| VegCA | Cellular Automata algorithm to simulate spatial organisation of PFTs | Zhou et al. (2013) |
| PrecipitationDistribution | Generate a storm sequence of intervals and intensities | Eagleson (1978) |
| FireGenerator | Produces intervals between fire events, following a Weibull distribution | Polakow & Dunne (1999) |
| StreamPowerEroder | Implements fluvial erosion according to stream power, using the Fastscape algorithms | Braun & Willett (2013) |
| FastscapeEroder | An alternative implementation of the Fastscape stream power algorithms | Braun & Willett (2013) |
| DetachmentLtdErosion | An implementation of stream power erosion *not* based on Fastscape | Howard (1994) |
| SedDepEroder | Sediment-flux dependent shear stress based fluvial incision | Hobley et al. (2011) |
| SteepnessFinder | Calculates steepness indices for a channel network | Wobus et al. (2006) |
| ChiFinder | Calculates the chi index along a channel network | Perron & Royden (2012) |



**Figure captions:**

**Figure 1: Examples of surface-process models.** (a) Computed depth-to-groundwater, from the tRibs flood forecasting model (Ivanov et al., 2004, image courtesy Enrique Vivoni). (b) Computed patterns of soil erosion and sedimentation on

agricultural fields, using the SIMWE soil erosion model (Mitas and Mitasova, 1998). (c) Model of ice-age glacier extent over the Sierra Nevada Mountains, using the GC2D iceflow model (Kessler et al., 2006). (d) Simulation of canyon erosion and fan-delta progradation in a region of active uplift (top) and subsidence (bottom), using the CHILD landscape evolution model (Tucker and Hancock, 2010). (e) Model of simultaneous cratering and fluvial erosion on the ancient Mars surface, with the MARSSIM model (Howard, 2007). (f) Simulation of pyroclastic flows at Tungurahua volcano, Ecuador, using the

VolcFlow model (Kelfoun et al., 2009).

**Figure 2: Schematic illustration of the structure of Landlab 1.0.** The three main divisions of the code are the grid, the components, and supporting utilities. Structure within these three main divisions is discussed in the main text.

**Figure 3: Geometry and topology of grid elements on various Landlab grids.** Only one patch and its bounding links are shown for each example to prevent the diagram from becoming cluttered. Links always point into the upper right semicircle, as described in the text.

**Figure 4: Standard ordering schemes and conventions in Landlab.** Examples are shown for both a small

RasterModelGrid (a) and a small VoronoiDelaunayGrid (b). Point elements (nodes, corners) are numbered in black plain text, areas (patches, cells) in black italics, and linear elements (links, faces) in gray italics. Symbols are as in Fig. 3. In all grid types, elements are ordered by y then x according to their geometric centers. Directional elements (links, faces) always point towards the top right quadrant. Rotational ordering is always counter-clockwise from the positive x-axis (right/east). This includes angle measurements. Examples of calls to grid properties are shown alongside each grid type to illustrate the

expression of these ordering rules in practice. Note that corners, faces, and cells are not shown in (b) for clarity.

**Figure 5: Interplay of node and link boundary conditions on a Landlab example grid.** Because nodes rather than corners define the outer margin of the grid structure, the perimeter nodes lack cells, and the perimeter links lack faces (see main text). These aberrant nodes and links are automatically set as boundary elements. Landlab defaults to setting the

condition of any such node to FIXED_VALUE_BOUNDARY and any such link to INACTIVE.

**Figure 6: Interaction of timescales between a Landlab driver and a set of components.** In this example, a driver that implements components 1–4 has a time loop of length *dt*, and *dt* is the timescale that is passed to the components. Components 1 and 2 implement numerical schemes that have maximum stable timesteps shorter than *dt*. In these cases, the





imposed *dt* interval is internally subdivided to ensure the model remains stable. Here, we see two possible ways a component might do this, either always taking the largest timestep possible then a short timestep to finish (component 1), or by dividing the imposed timestep into the minimum number of equal length internal steps, $dt_{int}$, where $dt_{int} < dt_{stable}$ (component 2). Even if a component could run for a timestep longer than *dt* (e.g., components 3 and 4), under an explicit-time Landlab driver

script like this, its steps will be truncated at *dt*. Once all the components have run for *dt*, they sequentially update their output fields in the grid with their changes. This is the only time that information can be passed actively between each component (and the driving script, if it also makes changes to the grid fields within the loop); each component cannot "feel" changes being made by any other until *dt* has elapsed. Hence *dt* is best thought of as the "coupling timescale".

**Figure 7: Schematic illustration showing how Landlab's grid geometry may be used to construct a finite-volume numerical scheme.** White squares represent nodes, with example node IDs given for a 5x5 raster grid. Gray ovals show the centre points of the links, with the link IDs given. In this example, we assume that we have a node field called "elev" whose values represent the altitude of the land surface at various node locations (example values shown in italics next to each node). Black arrows indicate direction of soil flow (in the down-hill direction). A finite-volume solution for a diffusion

model can be implemented by (1) calculating the gradient at each pair of adjacent nodes and assigning it to the corresponding link (lines 1–3 in the code snippet below), (2) multiplying by a transport-rate coefficient (and -1) to obtain unit flux (lines 4–6), and (3) multiplying the unit flux at each cell face by the width of that face, and adding up the inflows and outflows, and dividing by cell area to obtain flux divergence (lines 7 and 8).

**Figure 8: A simple finite-volume hillslope diffusion model implemented in Landlab.** Values adopted here are within typical terrestrial ranges for hillslope length (~100 m, controlled from line 3), hillslope diffusivity (0.01 m² y⁻¹, line 6) (Fernandes and Dietrich, 1997), total time of run (around a million years, since *dt* ~ 1833 y, lines 7-8), and uplift rate relative to baselevel (0.001 m y⁻¹, line 9).

**Figure 9: Hillslope diffusion implemented in Landlab using a component.** Compare to Fig. 8. Note that this version is more concise, and that timestep stability is now handled internally within the component.

**Figure 10: Performance of a Landlab-built model of landform evolution, using the StreamPowerEroder, FlowRouter, and PrecipitationDistribution components on grids of different types and sizes.** Runs were performed on a Mid-2014

Macbook Pro, and each data point represents the mean of five runs. (a) Total time for a simulation of 3 million years, implementing a stochastic storm sequence of around 3000 distinct stormy intervals. Both the total time to run and the time spent in the loop in the code that iterates forward in time are shown, and are practically indistinguishable in most cases. The time to run the components is close to linear with number of nodes, as expected for the Fastscape algorithms (see main text). (b) The time spent initializing the grids and components in each case (i.e., the total time less the time spent in loop from (a)).



Setting up a Voronoi-based grid is more computationally expensive than a raster, but both are quick in absolute terms, and both are close to linearly scaled with the number of nodes. In both graphs, small deviations from linear scaling occur, probably related to the interaction of Python's dynamic memory management with the size of the random access memory on the individual machine.

**Figure 11: Simulated topographies produced from a simple stream power-based fluvial incision rule, combining the StreamPowerEroder, FlowRouter, and PrecipitationDistribution components.** The same model set up is implemented on both a RasterModelGrid (a) and a HexModelGrid (b), using the same random seed to generate the topography. Note the vertical-horizontal asymmetry in channel network planform visible in (b), an expected outcome of the three axes of mirror

symmetry running though a hexagonal grid. The linearity of these catchment planforms is enhanced by the presence of an erosion threshold.

**Figure 12: Simulated topographies produced from a coupled hillslope and channel evolution model, combining the StreamPowerEroder, FlowRouter, and LinearDiffuser components.** A storm sequence is provided by the

PrecipitationDistribution component, and discharge is routed across depressions in topography using DepressionFinderAndRouter. Stream-power parameters are identical to those in Fig. 11. The same model setup is implemented on both a RasterModelGrid (a) and a HexModelGrid (b), using the same random seed to generate the topography. Despite the differences in grid organization, planform drainage pattern remains fairly similar between the two cases.

**Figure 13: Example of a CellLab-CTS model.** Here the CellLab-CTS framework has been used to implement a model of granular mechanics. The model has eight node states, representing air (white), a resting grain (light grey), and a grain moving in each of the six lattice directions (all coded as dark grey). Grid edges and immobile walls are treated as CLOSED_BOUNDARY Landlab boundary conditions (black). Transition rules are used to model grain motion, grain collision,

and gravity (from Tucker et al., 2016).

**Figure 14: Implementation of an ecohydrology model in Landlab.** (a) Schematic illustration of coupling among different Landlab components for the CaTGraSS application. (b) Demonstration of the model on a flat surface with semi-arid climate similar to that of Sevilleta, New Mexico, USA (Zhou et al., 2013). This figure plots percentage of space occupied by each

PFT with time. (c) Spatial organization of PFTs at different times during the model run. These plots illustrate competition between different PFTs for space and resources. Trees die early within the first 300 years due to unfavorable climatic conditions and competition from shrubs and grass. The ecosystem swings between shrub-dominant and grass dominant states for the next 1600 years.





**Figure 15: Demonstration of OverlandFlow component capabilities**. The example shows development of a hydrograph in a catchment drawn from an airborne Lidar-derived DEM of the Spring Creek catchment in central Colorado, USA. The run uses a constant rainfall rate of 25 mm h$^{-1}$ and a storm duration of 1 h. The hydrograph persists for almost 8 model hours, and water depth as plotted at several intervals after the start of the precipitation

5    event: 1 h (the end of the storm), 2 h, 3 h, and 8 h.





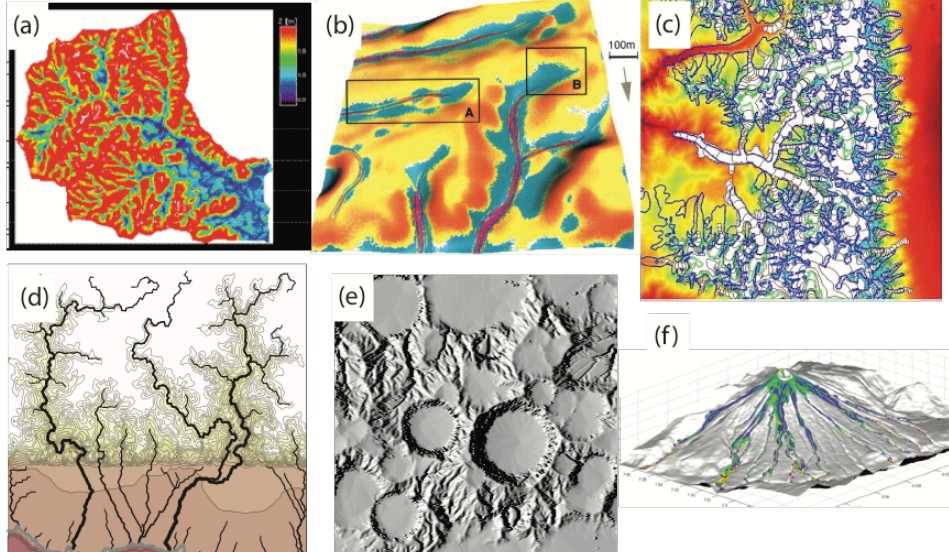

Figure 1




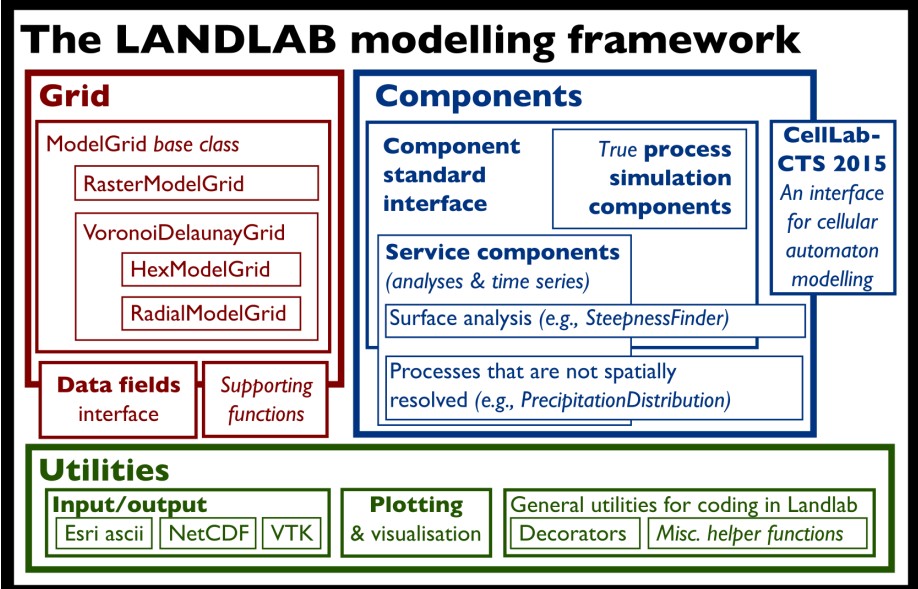

Figure 2



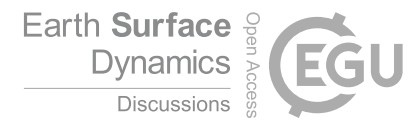

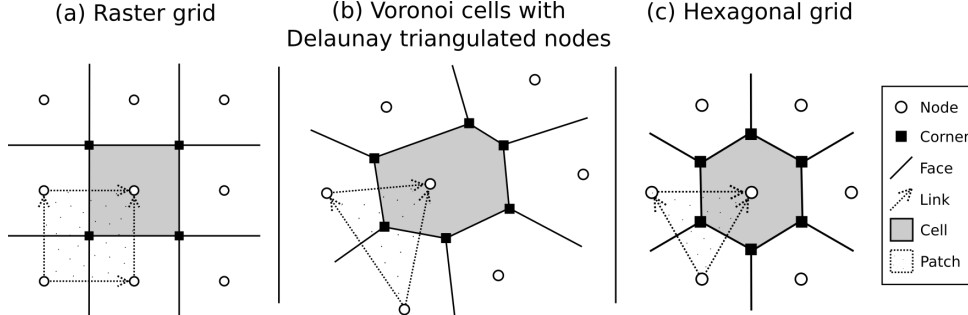

Figure 3





(a) Raster grid
*(i) nodes, links, and patches*

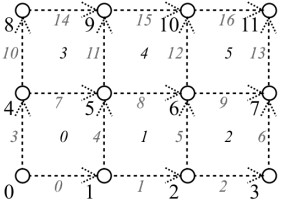

```
raster.links_at_node[[6, 7]]
          = [[  9,  12,   8,   5],
             [ -1,  13,   9,   6]]

raster.links_at_patch[0]
          = [ 4,  7,  3,  0]
```

*(ii) corners, faces, and cells*

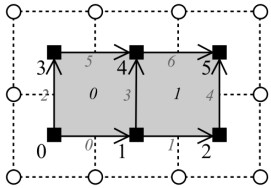

```
raster.faces_at_cell
          = [[3, 5, 2, 0],
             [4, 6, 3, 1]]
```

(b) Voronoi grid

*(i) nodes and links*   *(ii) link directions and patches*

(c) Grid ordering and directional conventions

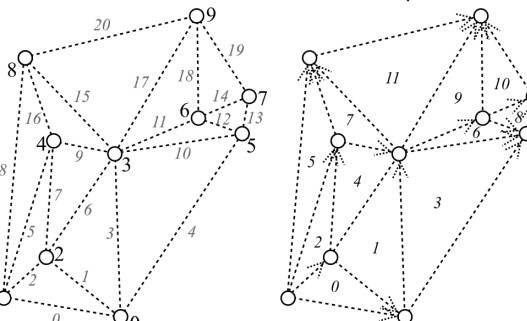

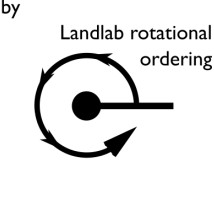

Elements ordered by y, then x

Landlab rotational ordering

positive link orientation

```
voronoi.neighbors_at_node[3:6] = [[ 5,  6,  9,  8,  4,  2,  0],
                                  [ 8,  1,  2,  3, -1, -1, -1],
                                  [ 7,  6,  3,  0, -1, -1, -1]]

voronoi.angle_of_link[[0, 1, 2]] = [6.0974, 5.2275, 1.3141]
```

Figure 4



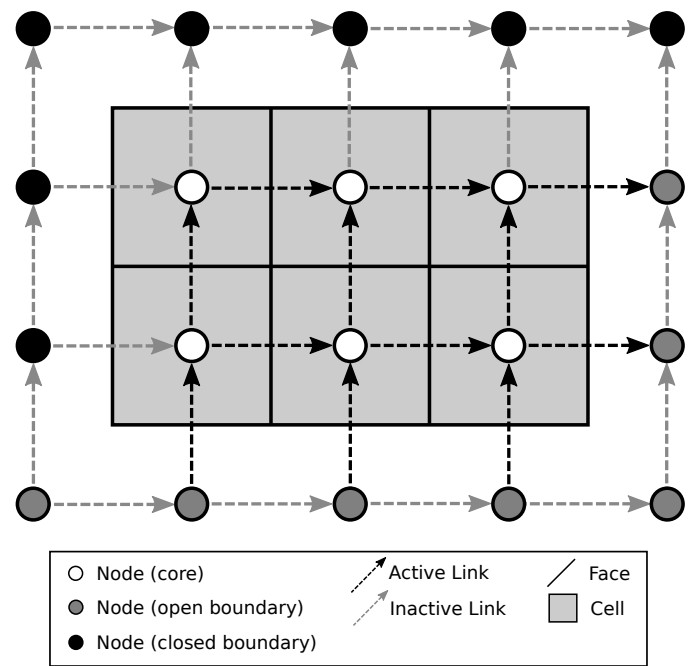

Figure 5



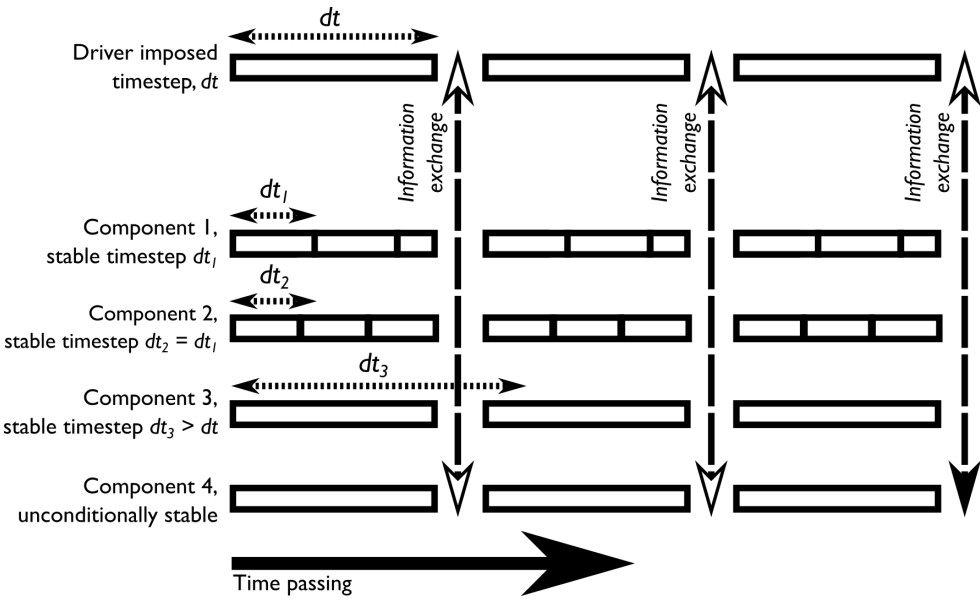

Figure 6





Figure 7





*Code to implement a simple diffusion model on a radial Landlab grid:*

```
 1. >>> from landlab import RadialModelGrid, imshow_grid
 2. >>> from matplotlib.pyplot import show
 3. >>> mg = RadialModelGrid(num_shells=10, dr=10.)
 4. >>> z = mg.zeros('node')
 5. >>> qs = mg.zeros('link')
 6. >>> diffusivity = 1.e-2
 7. >>> dt = 0.2 * mg.length_of_link.min() ** 2. / diffusivity
 8. >>> for i in range(500):
 9. ...     z[mg.core_nodes] += 0.001*dt
10. ...     g = mg.calc_grad_at_link(z)
11. ...     qs[mg.active_links] = -diffusivity * g[mg.active_links]
12. ...     dqsdx = mg.calc_flux_div_at_node(qs)
13. ...     dzdt = -dqsdx
14. ...     z[mg.core_nodes] += dzdt[mg.core_nodes] * dt
15. >>> imshow_grid(mg, z, grid_units=('m', 'm'), var_name='Elevation (m)')
16. >>> show()
```

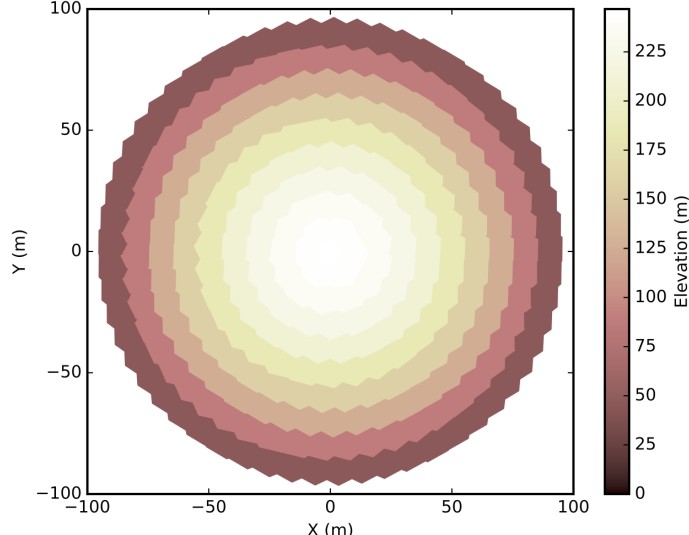

Figure 8




*Code to implement a simple diffusion model on a radial Landlab grid, using Landlab components:*

```
1. >>> from landlab import RadialModelGrid, imshow_grid
2. >>> from landlab.components import LinearDiffuser
3. >>> from matplotlib.pyplot import show
4. >>> mg = RadialModelGrid(num_shells=10, dr=10.)
5. >>> z = mg.add_zeros('node', 'topographic__elevation')
6. >>> dt = 2000.  # no longer the stable timestep
7. >>> ld = LinearDiffuser(mg, linear_diffusivity=1.e-2)
8. >>> for i in range(500):
9. ...     z[mg.core_nodes] += 0.001*dt
10. ...     ld.run_one_step(dt)
11. >>> imshow_grid(mg, z, grid_units=('m', 'm'), var_name='Elevation (m)')
12. >>> show()
```

Figure 9





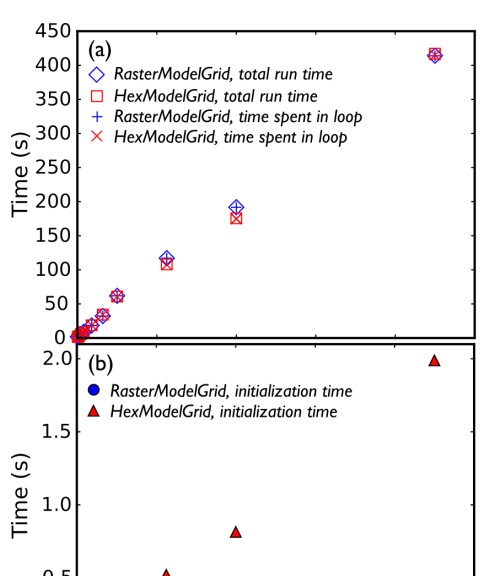

Figure 10



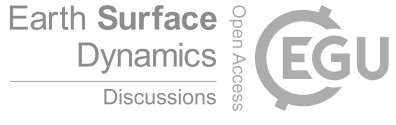

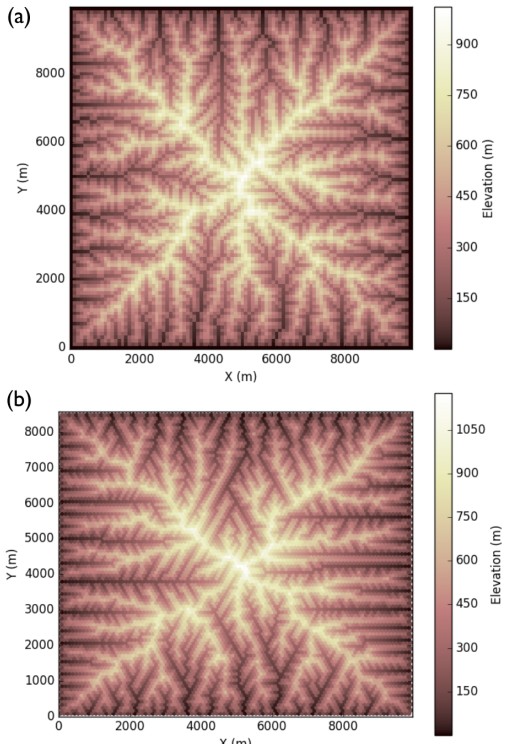

Figure 11

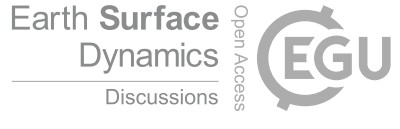

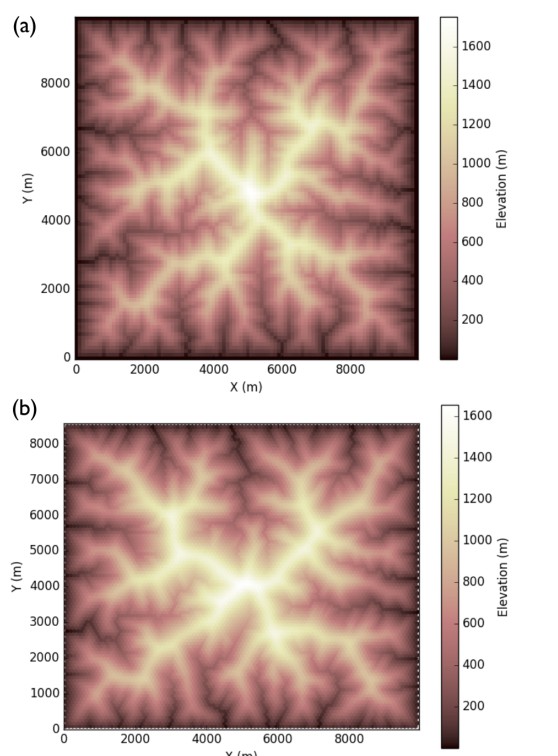

Figure 12


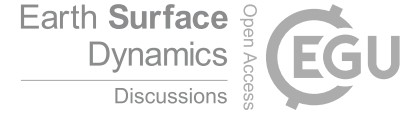

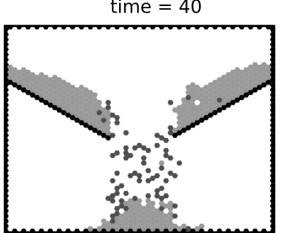 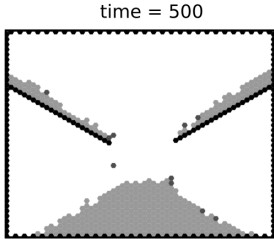 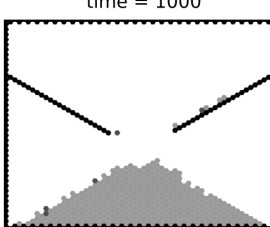

Figure 13



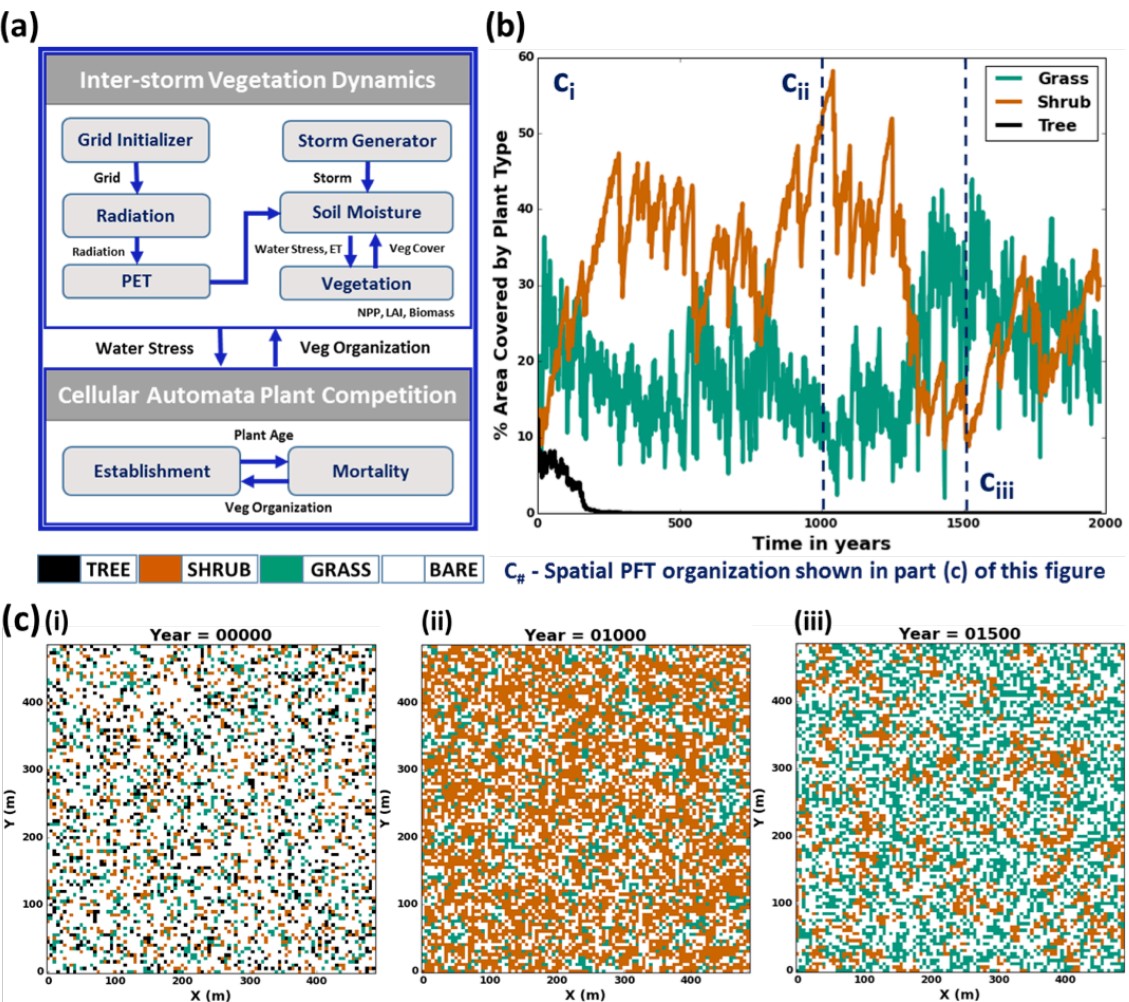

Figure 14





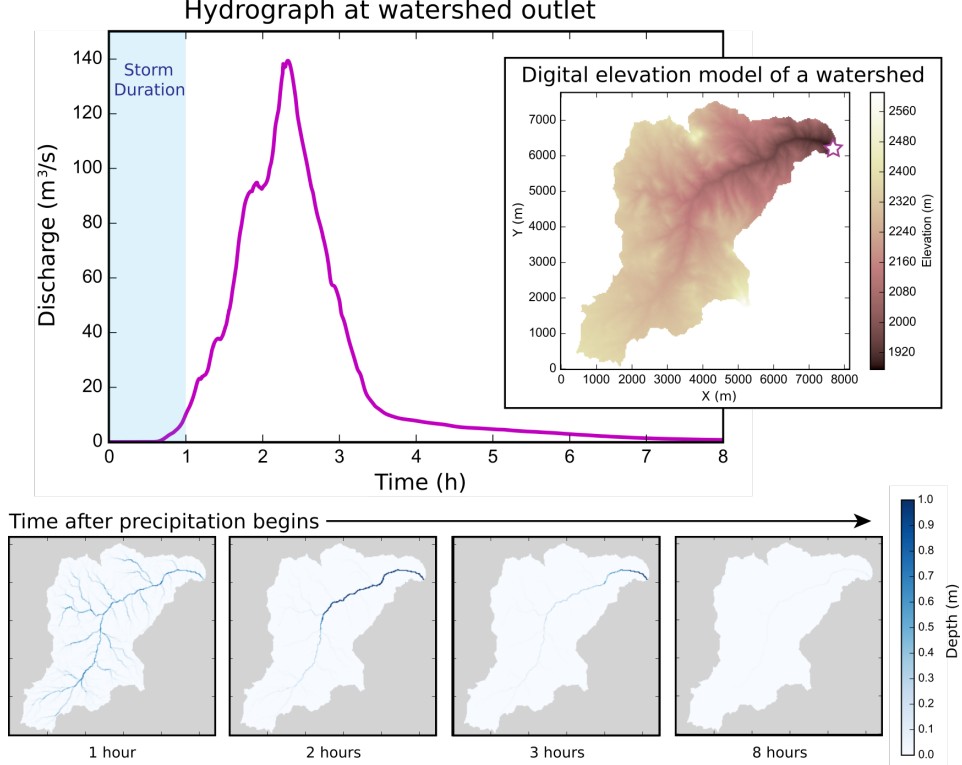

Figure 15