# Peer review of "Creative computing with Landlab: an open-source toolkit for building, coupling, and exploring two-dimensional numerical models of Earth-surface dynamics"

_Earth Surface Dynamics, 2016_

## Short Comment (SC1) · 5 Oct 2016

I congratulate the authors to provide the community with Landlab and this accompanying paper. As the authors share their model as an entirely open software on GitHub, they adhere to the growing trend of open collaboration for scientific computing. This is great! This being said, I found it a bit surprising to find TopoToolbox and TecDEM referenced as closed source software (page 4, line 9), in line with ArcMap and Matlab. I agree that ArcMap and Matlab are closed and commercial software. Yet, the Matlab-based toolboxes TopoToolbox and TecDEM themselves are open software as

their codes are open and could be run (although with major modifications) on platforms like Octave, which itself is open. I think that there is a smooth transition between fully open software such as Landlab which is based on Python, software such as TopoToolbox and TecDEM that exploit the increasing openness and interoperability of commercial and closed software such as ArcGIS and Matlab, and completely closed-source software. I encourage the authors to account for this slight but significant difference in principles of software development.

Wolfgang Schwanghart

––––––––––––––––––––––––––––––––––

---

## Author Comment (AC1) · 6 Oct 2016

Dear Wolfgang,

Your comments are well-taken. We will work harder in the forthcoming revision of the manuscript to emphasise the spectrum of accessibility of existing codes, particularly re Topotoolbox and TecDEM. The status of codes which themselves are open source but nonetheless exist as extensions to major pieces of closed-source software is interesting, as you point out, and certainly worthy of emphasis here. We'll ensure the revision adds this nuance.

[Figure]

Thanks for taking the time to read the manuscript and for sharing your thoughts.

Dan Hobley
* * *
**ESurfD**

---

## Referee Comment (RC1) · A. Wickert (Referee) · 24 Oct 2016

**Summary**

Hobley and co-authors present Landlab, a toolkit that is poised to significantly change the face of Earth-surface modeling. I expect their framework to easily build and integrate models of the evolving Earth's surface to enable the community to improve – possibly rapidly – the state of our knowledge and theory of Earth-surface dynamics. On a personal note, I have followed the development of Landlab since its inception,

and offer my enthusiastic support of the publication of this article that makes this comprehensive modeling infrastructure easy to follow in a way that mirrors the clean and well-written code base.

I recommend that the article be accepted for publication after some very minor technical corrections. As the article is, in my opinion, ready for publication, this review contains both technical notes and copyedits as a result of a very thorough read.

**Main text**

General: Many geomorphic models are run in 1 dimension. Is Landlab capable of solving 1-D problems? As I understand from my reading, it is designed for 2-D solutions, but it would be helpful to have a comment on this to clarify.

Page 4, line 9: I second Wolfgang Schwanghart's comment regarding open-source modules written in closed-source languages. Here, the algorithm may be known even if it cannot be run without the closed-source core programming language.

Page 6, line 9: Strong agreement regarding rapid prototyping. Based on reading the article, and having never used Landlab before (though having some familiarity with it), I was able to assemble a simple program for a class exercise in 1-2 hours.

Page 9, lines 6-10: Easy to note that this is standard right-hand-rule, if you think this is helpful.

Page 10, *"get" and "create" methods*: I have seen "get" commands, such as "mg.get_grid_ydimension", that are exposed to the end user and have no underscore. So I must guess that there are two types of "get" commands, or... ?

Page 11, section 3.1.4: More a comment: the single set of boundaries implies to me

**ESurfD**
that you have intended the model for discretization of only first and second derivatives.

Page 12, line 16: Missing a noun between "ID" and "of".

Page 12, line 24: "Pairwise Transition Automata" appears, on Google and Google Scholar, only in this article, and its meaning is unknown to me. Could you elaborate?

Page 13, line 20: What is "syntactic sugar"?

Page 13, line 28: Missing hyphen for steady-state (modifies "solutions")

Page 15, line 18: Suggest replacing "However" with "While these are focused on Earth's surface" (or "While these are Earth-surface focused", if you like some dislike using the vestigial English genitive in scientific writing.

Page 16, line 4: Having not thought about it before, it is not clear to me how you would combine a set of Jacobian matrices to, presumably, simultaneously solve an arbitrary number of coupled processes. It is probably just my ignorance, but I would like to see a bit of explanation and/or a reference.

Page 16, line 9: changes → change; the word "data" is plural.

Page 17, line 19: Are "known correct solutions" all analytical solutions, or do you include solutions that go beyond this limited class of solutions? And/or, is "solutions" meant more generally – as in, is this also just making sure that general functions that you have programmed that may not be generating mathematical solutions are functioning as expected? As I type, I'm starting to think it must be the latter...

Page 18, line 26: uplifting → uplift

Page 20, line 15: More general question: Your method implies central differencing.

How do you ensure that numerical diffusion is minimized and/or is not large enough to be important?

Page 21, line 8: Would this be the depth-integrated flux, as "flux" generally means [quantity/[area time]], so volumetric flux is L/t and therefore depth-integrated would become $L^2/t$? Of course, Earth-system modelers use "flux" more liberally, but in a mathematics (and fluid mechanics), this is the use that I know, and I think it's best to be precise with language – especially considering how much you are in general.

Page 21, line 11: Again, more general question: if you are using the CFL condition, I am guessing that this is referring to something that is Euler forward. How would you choose time steps internally while using some of your implicit methods?

Page 22, Equation 9. Note $\geq$ is possible ($\geq$) in LaTeX. There are also options to use curly brackets to make the conditional part of the equation clearer; see http://tex.stackexchange.com/questions/47170/how-to-write-conditional-equations-with-one-sided-curly-brackets.

Page 22, line 22. Near $m/n$, you may mention that this is the channel concavity. Perhaps, "$m/n$, i.e., channel concavity, $\approx 0.5$.

Page 22, line 26. Remove "possible" (repeated later)

Page 23, line 2: Remove "performs as order-n, and as expected": redundant. You could keep "as expected" if you like.

Page 23, lines 3-4: "Broadly linearly" sounds strange: two dimension terms, the first metaphorical. How about "approximately", or even just no modifier (it is so close to linear)?

Page 23, lines 17-19: "based on...2001b)." is a repeat of text at the start of section 5.2

(p. 22, l. 2-6).

Page 23, line 20: "storm depth" to "stormwater depth" (or something like this)?

Page 23, lines 24-25: "looped upon" to "called within the loop", and remove "until...cease". If I understand you here, this is a simpler way of writing this.

Page 23, line 28: "and where it only rains" to "with rainfall occurring" for better parallel structure

Page 25, line 5: remove comma after "region"

Page 25, line 6: remove "using" after "by": unnecessary

Page 25, line 6: did you just write that stormwater depth is given by a gamma function? Also, "storm depth" should be turned into something like "stormwater depth" that makes more sense, per the above comment.

Page 25, line 9: Does Bras' model incorporate latitude, integration over a day, etc.? And is it simple incident radiation (i.e. no diffuse or reflected)? I think that the latter is true from what you have written, but not sure about the former.

Page 25, line 21: "and also" → "and"

Page 25, line 29: "shrubs cluster as they..."

Page 26, line 12: "shallow" is ambiguous since you are writing about flow depth and slopes. I would write "low-slope" instead. Also, did you mean a different word than "urban"?

Page 26, line 29: "a uniform rainfall rate". Also, this is by filling the pits with water, I

presume.

Page 28, line 4: You haven't mentioned visualization except for one reference to Par-aView that you didn't follow up and a passing reference to matplotlib. Not actually a problem, but a note.

**Tables**

Table 1b: What about raster D8?

Table 2: Should rectilinear have rectangular cells, not "quasi-rectangular"?

Table 4: More of a comment: based on all boundaries being inactive, a 1D model in Landlab would have to have $3 \times (N + 2)$ nodes. So it really isn't optimized for 1D, it seems.

Table 5: A nice set of modules... but at risk of sounding ungrateful of the large amount of work that this was, it would be really great to see a depositional components for Landlab appear... in case this isn't obvious, this is not a criticism of the paper, but more a comment on where this work could lead.

**Figures**

Figure 3: Please ask copyeditors to ensure that this appears on the same page as Table 1.

Figure 4: Are patches numbered in order of min(max(surrounding nodes or links))? (Question inspired by (b)(ii).)

Figure 7: Include line numbers. Also, flux → discharge, per our discussions? In addition, for this and the other figures with code, I am not sure if »> and . . . are needed or are distracting... they are more the latter for me, I guess. But your decision in the end on style.

Figure 8: You use both "base level" and "baselevel"; check for consistency (2x/each).

Figure 9: Why not combine this with Figure 8, if they make the same plot?

Figure 10: Please tell us how many cells you computed over to help readers gauge their own compute times.

Figure 14: (b) Vertical axis: PFT is more precise than plant type? Although this is clear from everything else. Legend: shown in parts (b) and (c) of this figure.

**Supplement**

No concerns.

**Testing**

During the review process, I took advantage of the fact that I am teaching a geomorphology class to test Landlab. In our modeling lab, the students, most of whom have little experience in programming, were all able to install and run Landlab. In some cases, differences between computers and Python versions caused a little confusion, but the code itself worked well. The students are currently working using it with a simple coupled channel–hillslope model to (1) find relationships between hillslope diffusivity, the stream power coefficient, and the drainage area of the hillslope–channel transition, (2) investigate which factors lead to changes in the time it takes to reach

steady-state conditions in a landscape, and (3) explore the effects of changing grids, uplift patterns, stream-power exponents, and more on the model results. The overall reception was very positive in this mixed undergraduate–graduate class, with students realizing through the model how real landscapes form and evolve. The code that I used, modified from one of those supplied in the paper, is attached below. I intend to teach the lab again.

One issue that I had while running this code on Linux within iPython is that the plotting tools would not release control of the shell to me (i.e. after plotting, I could not type anything more into the terminal, even after closing all of the graphical windows). This of course is not an issue to preclude publication, but I will add it as a ticket on GitHub.

---

## Referee Comment (RC2) · B. Campforts (Referee) · 11 Nov 2016

**Review of Hobley et al. creative computing with Landlab**

With Landlab, Hobley et al. present a refreshing framework to integrate numerical solutions for planetary surface processes. The model is fully open source and targets both end-users and developers providing an extended library of model structures and implementation strategies. The wealth in existing models and model structures makes the paper timely for the geophysical earth surface community and I recommend the paper for publication after consideration of some remarks.

Before reading the paper and because of its intention: i.e. providing an integrative numerical modeling toolkit, I formulated some questions which I would like to see addressed in such a paper. In the following I list those questions and discuss to what extend I found satisfying answers and where the paper can be improved.

- General ease of use of the numerical model.

I congratulate the authors for their well-functioning and carefully documented source codes.

- Debugging tools

I highly value the availability of unit tests to evaluate the model code. This is one of the aspects in which Landlab differentiates from other numerical earth surface models.

- Input/output data structures. Compatibility with other tools available.

Well implemented and properly discussed in the paper

- Discretization of the continuous solution to grids. Shifting between different kind of grids.

Another major contribution of the paper is the way in which different discretization schemes and the numerical grids which come with them are presented. Nonetheless, while reading section 3.1, I sometimes had the feeling reading technical notes rather than a paper on earth surface dynamics. I propose that the authors discuss, somewhere in this section, the importance of different grid implementations and its relevance for the field of earth surface dynamics. E.g. the use of structured grids has major implications in terms of landscape symmetry as elegantly documented by Braun and Sambridge (1997). It would also be interesting to briefly discuss the implications of using raster versus voronoi grids and to what extent the model supports conversion between the methods. A first example which comes to mind in the framework of LEMs (which is only partly what the model is designed for) is the simulation of tectonic shortening which is often executed on irregular grids (Willett, 1999), whereas other landscape processes such as nonlinear diffusion (Perron, 2011) might benefit from structured grids for computational performance. Does the grid structure of the model allows for grid refinement (e.g. Künze and Lunati, 2012)?

- Parallelization and suitability for supercomputing

One essential asset of a robust model structure, especially if the authors target a large user community, is the possibility to port the source code to a larger computer infrastructure. This item is currently not covered in the paper and it would be good to discuss on the potential of Landlab to be parallelized and the limitations which comes with it. In case the model supports parallelization, how do you take care for changing drainage areas while executing the model on different computational blocks?

- 3 spatial dimensions

Many earth surface processes require a 3-dimensional spatial discretization. Amongst others: hydrological processes, ice dynamics, terrestrial heat advection and diffusion, and soil dynamics. As the authors explicitly state that Landlab *"is not a landscape evolution model …rather, it presents a framework under which a wide variety of models can be implemented using it tools, including hydrologic …",* I am a little surprised that the use of a third spatial dimension is never mentioned in the text. I do not suggest that the $3^{rd}$ dimension should be included in the current release of the software but I would find it interesting to see a discussion on the possibility of adding a third spatial dimension to the model.

- Numerical accuracy.

During last decades, many numerical models have been developed mainly focusing on (i) earth surface processes being simulated or (ii) the performance of the numerical simulations. Much less attention has been given to the numerical accuracy of the developed models. This issue is also not very well covered in this manuscript. Nonetheless, recent work has shown that numerical accuracy significantly influences model performance, not only when topographical knickpoints are present but also when lateral displacement of topography needs to be accounted for (Campforts et al., 2016; Campforts and Govers, 2015). Moreover, it would be good that Landlab also offers some analytical solutions for the numerical processes being simulated or that the authors discuss how this could be achieved as analytical solutions are the benchmark against which different solution strategies can be evaluated. As such, a benchmark solution would be available which cannot only serve as a tool to evaluate the accuracy of the current model structure but is also of aid to new users in order to test new models and numerical solutions. For example, an analytical solution for the stream power model can be quite easily found using the slope patch solution of Royden and Perron (2013) but also for numerical solutions not directly related to landscape evolution, analytical solutions do exist (Stüwe et al., 1994). Numerical accuracy is sometimes mentioned throughout the text (e.g. when discussing the influence of timesteps on line 30, p. 15) but it would be good to elaborate on this point and cover this relevant issue in more detail

- To what extend is the model oriented towards integration of measured data into the numerical model?

Major advances can be made in the field of earth surface processes by combining numerical models with field data (Fox et al., 2015; Glotzbach, 2015). I think the framework presented in this paper offers an excellent opportunity for such an approach and could be of high value for otherwise more data oriented researchers. It would be interesting to see some explicit suggestions or remarks on how to combine the Landlab framework with field data.

- Particle tracking

One very interesting aspect of numerical models is to trace back particles through time. I am wondering whether Landlab allows fingerprinting of for example sediments. Is it possible to track a sediment particle all the way from the hillslopes to the outlet of the drainage network? Can this be done for all particles and how feasible is that in terms of memory allocation and data storage? Particle tracking would offer great potential for the model to be further combined with e.g. detrital dating methods (thermochron data (Herman et al., 2015)/ CRN data (Mudd, 2016)) ).

- An interoperability interface

In the abstract, Landlab is presented as a modular framework with a strong interoperability. The paper comes up to this promise and presents four contrasting model designs. I enjoyed reading these sections and I think they strongly contribute to the illustrative nature of the paper. Nonetheless, I think this section could be even stronger if the authors also point out the advantage having the different model approaches under the roof of one single model infrastructure. In my opinion, one of the main assets of Landlab is not necessarily the capacity to execute separate model designs but exactly that they can be combined. For example, until now, little attention has been given to the role of vegetation in landscape evolution models. Nonetheless, it has been shown that vegetation might have a very strong influence on landscape evolution (Collins, 2004). The cellular automaton presented in sections 5.3 could therefore be neatly integrated with section 5.1 and/or 5.2. Likewise, hydrological processes are shown to govern stochastic processes like shallow landsliding (Montgomery and Dietrich, 1994). Hence section 5.4 can be perfectly linked up with the other sections. I would keep the sections in their current form as they are very illustrative of the model performance but it would be definitely nice to discuss and eventually illustrate the potential to integrate all or some of these modules.

**Minor comments**

Overall the paper is very fluently written and I enjoyed reading it. I endorse the comments of W. Schwanghart and A Wickert. In addition, I have a few minor comments:

Page 2-5: Introduction. I get the point the authors try to make but the way the introduction is written feels like the authors want to promote their product as the one and single framework to be used for future model development. Although I agree with the authors that the structure they provide can offer excellent guidance for new model development and the 'gridding engine' can definitely save many hours of numerical coding headache, I am not so sure whether I agree it is a good idea to demotivate building code from scratch. The latter is still the best way to become familiar with the implications and limitations which come with numerical software design and to realize to power and caveats of numerical models. I second W. Schwanghart's comment in suggesting that many open source initiatives exist, although sometimes written in commercial software (e.g. TopoToolbox and the LEM TTLEM is fully open access; SIGNUM (Refice et al., 2012)) or even in Python, advanced GIS packages are available (LSDTopoTools https://github.com/LSDtopotools).

Page 10, line 24: Analytical method. I would use a different wording as analytical typically refers to the analytical solution of e.g. PDE's. What about *grid methods*?

Page 15-16, line 23-5: I find this very interesting. Does the model allow for constraints on the timestep in the case that implicit methods are being used? Although they are indeed unconditionally stable, the use of large (main model) timesteps in LEMs might result in in very sudden topographical changes causing the presence of artificial steps in the landscape. E.g using an uplift rate of 1mm/yr in combination with timesteps of 10ky results in a sudden uplift of 10m in the LEM which initiates artificial knickpoints at the baselevel of rivers.

Page 18-19: nice illustration on the use of finite volume methods.

Page 22, line 31: what about maximum timesteps for implicit schemes? See comment above (Page 15-16)

Page 24 line 22: I would get rid of the sentence starting with Figure 13. I also do not see the additional value of Fig 13 for this paper.

**Figures**

Great figures in general.

Figure 1: Nice images but I find it a missed opportunity to illustrate the broad range of problems Landlab itself is capable to simulate.

Not sure what the additional value of the code snippets is. Personally, I find the Landlab wiki page much more insightful in this perspective.

**Testing:**

I am not very familiar with python and I second the comments of referee 1 who did elaborate testing with a large user group. Nevertheless, I installed the software and executed some of the tutorials which I found very easy to understand and clearly documented. I had no single problem when executing the code. A very small suggestion would be to add a link to the 'coupled_params_storms.txt' file in the 'Getting to know the Landlab component library' tutorial on https://nbviewer.jupyter.org/github/landlab/tutorials/blob/master/component_tutorial/component_tutorial.ipynb, similar to the link given for the 'coupled_params.txt' file.

---

## Author Comment (AC2) · 24 Nov 2016

We have chosen to remove the reference to Topotoolbox and TecDEM to remove the chance that a reader could misinterpret the status of these codes, which as Wolfgang notes, is not as black-and-white as their inclusion in this section suggested.

---

## Author Comment (AC3) · 24 Nov 2016

Dear Andy,

Thanks for taking the time to complete such a thorough review. As requested by the AE, please find below our replies to your comments. We've pasted each comment you made, then replied to it, and as necessary then copied in any modifications made in the text. We hope that these replies address your concerns to your satisfaction.

Thanks again,

[Figure]

Dan (on behalf of all the authors)

Summary

Hobley and co-authors present Landlab, a toolkit that is poised to significantly change the face of Earth-surface modeling. I expect their framework to easily build and integrate models of the evolving Earth's surface to enable the community to improve – possibly rapidly – the state of our knowledge and theory of Earth-surface dynamics. On a personal note, I have followed the development of Landlab since its inception, and offer my enthusiastic support of the publication of this article that makes this comprehensive modeling infrastructure easy to follow in a way that mirrors the clean and well-written code base.

I recommend that the article be accepted for publication after some very minor technical corrections. As the article is, in my opinion, ready for publication, this review contains both technical notes and copyedits as a result of a very thorough read.

Main text

General: Many geomorphic models are run in 1 dimension. Is Landlab capable of solving 1-D problems? As I understand from my reading, it is designed for 2-D solutions, but it would be helpful to have a comment on this to clarify.

$\nu$ Added to grid section: "Although Landlab grids are inherently two-dimensional, in many cases it is nonetheless possible to create an effectively one-dimensional simulation by creating a 3-by-N regular grid and closing the nodes along the top and bottom edges (see Section 3.1.4). Three-dimensional grids are not possible in Landlab at this time, though may be supported in a future release."

Page 4, line 9: I second Wolfgang Schwanghart's comment regarding open-source modules written in closed-source languages. Here, the algorithm may be known even if it cannot be run without the closed-source core programming language. $\nu$ We've simplified this section by removing reference to these 'open-source-but-on-a-closedsource-framework' type pieces of software, as the level of detail required to do the subject justice seemed a bit much for what is ultimately a tangent to the manuscript.

Page 6, line 9: Strong agreement regarding rapid prototyping. Based on reading the article, and having never used Landlab before (though having some familiarity with it), I was able to assemble a simple program for a class exercise in 1-2 hours. $\nu$ DH: Glad to hear we're achieving our targets in this area...!

Page 9, lines 6-10: Easy to note that this is standard right-hand-rule, if you think this is helpful. $\nu$ DH: Added: "...(i.e., the right-hand rule)...."

Page 10, "get" and "create" methods: I have seen "get" commands, such as "mg.get_grid_ydimension", that are exposed to the end user and have no underscore. So I must guess that there are two types of "get" commands, or... ? $\nu$ DH: Good catch. There are nine of these, and they are stragglers from before Landlab's internal style standardization. All have direct modernized equivalents, and they should be expunged in the final release of Landlab that goes alongside this paper.

Page 11, section 3.1.4: More a comment: the single set of boundaries implies to me that you have intended the model for discretization of only first and second derivatives. $\nu$ As you note, all current components operate only with first and second derivatives. However, it would be conceptually straightforward to create more boundary condition types as and when the need arises to accommodate higher order derivatives.

Page 12, line 16: Missing a noun between "ID" and "of". $\nu$ I don't think so; but nonetheless this comment indicates this phrasing isn't super clear. I've changed it a little to try to improve it: "By indexing these arrays with the IDs of element subsets, the values at specific locations and on each element type can be recovered."

Page 12, line 24: "Pairwise Transition Automata" appears, on Google and Google Scholar, only in this article, and its meaning is unknown to me. Could you elaborate? $\nu$ Modified to: 'For instance, pairwise transition automata (Narteau et al., 2001; 2009)

[Figure]

represent the states of cells on a grid as paired "doublets", with rules prescribed to govern the rates of transition between each doublet type. These are readily implemented in Landlab by mapping the pair states onto the links of a Landlab grid, and representing the corresponding automaton cell states at grid nodes (Tucker et al., 2016).'

Page 13, line 20: What is "syntactic sugar"? $\nu$ We now explicitly define this (rather delightful) piece of computer science jargon: "Landlab offers some degree of "syntactic sugar" for its field name interface – i.e., the field interface is made more user-friendly by the addition of more readable grid properties to query the fields at each element type, rather than requiring the user to access the both dictionaries directly. For instance, grid.at_node['my_field_name'] is equivalent to grid['node']['my_field_name']. In addition, Landlab also provides convenient shortcuts to create new fields of ones (grid.add_ones), zeros (grid.add_zeros), and from existing data (grid.add_field)."

Page 13, line 28: Missing hyphen for steady-state (modifies "solutions") $\nu$ Added, and similarly at ln 29.

Page 15, line 18: Suggest replacing "However" with "While these are focused on Earth's surface" (or "While these are Earth-surface focused", if you like some dislike using the vestigial English genitive in scientific writing. $\nu$ We've settled on: "Although these existing components are largely Earth-surface focused, we emphasize that Landlab permits modeling of the evolution of almost any two-dimensional system that lends itself to description by discretized systems of differential equations or cellular automaton rules."

Page 16, line 4: Having not thought about it before, it is not clear to me how you would combine a set of Jacobian matrices to, presumably, simultaneously solve an arbitrary number of coupled processes. It is probably just my ignorance, but I would like to see a bit of explanation and/or a reference. $\nu$ Given this remains entirely speculative material about future work, we've decided to remove this (admittedly pretty unclear) paragraph entirely, rather than invest the space needed to actually do this idea justice.

Page 16, line 9: changes → change; the word "data" is plural. $\nu$ Done

Page 17, line 19: Are "known correct solutions" all analytical solutions, or do you include solutions that go beyond this limited class of solutions? And/or, is "solutions" meant more generally – as in, is this also just making sure that general functions that you have programmed that may not be generating mathematical solutions are functioning as expected? As I type, I'm starting to think it must be the latter... $\nu$ Yep, the latter. I think this is strictly correct as written, and bringing up analytical solutions specifically implies that the unit tests are written only for the final solutions to differential equations in the components, which they are not. The grid is heavily tested too, for example. On this basis I don't intend to make changes here.

Page 18, line 26: uplifting → uplift $\nu$ For maximum clarity: "...or other similar scenario with a radially symmetric uplift field."

Page 20, line 15: More general question: Your method implies central differencing. How do you ensure that numerical diffusion is minimized and/or is not large enough to be important? $\nu$ This is indeed a central differencing method, as is widely deployed in handling diffusion problems in gridded models. However, our understanding is that numerical diffusion should only become an issue in advection-diffusion problems, or alternatively in upwinding schemes. This component as described here is neither, so a full discussion of the general issues around stability would seem a bit out of place at this point in the text (see also our reply to Campforts on numerical stability).

Page 21, line 8: Would this be the depth-integrated flux, as "flux" generally means [quantity/[area time]], so volumetric flux is L/t and therefore depth-integrated would become L2/t? Of course, Earth-system modelers use "flux" more liberally, but in a mathematics (and fluid mechanics), this is the use that I know, and I think it's best to be precise with language – especially considering how much you are in general. $\nu$ Indeed. We have added this wording.

Page 21, line 11: Again, more general question: if you are using the CFL condition, I

am guessing that this is referring to something that is Euler forward. How would you choose time steps internally while using some of your implicit methods? $\nu$ I'm not sure which implicit methods you're referring to here. As emphasized in section 3.3.2, each component handles its own timestepping internally, and independently from all other components (see also Fig. 6). Thus in this specific case, the LinearDiffuser component is using a timestep governed by the CFL condition, while a putative separate implicit component would be using its own (longer?) internal timestep as governed by some other stability condition. The "coupling timestep" set by the user (sensu 3.3.2) is different again. I note that issues of timestep stability also came up in Campfort's comments, which resulted in some new text (now on p. 15) described in reply to him. Hopefully this text will also clarify things in relation to your comment here.

Page 22, Equation 9. Note $\geq$ is possible ($\geq$) in LaTeX. There are also options to use curly brackets to make the conditional part of the equation clearer; see http://tex.stackexchange.com/questions/47170/ how-to-write-conditional-equations-with-one-sided-curly-brackets. $\nu$ This is not a LaTeX manuscript! This symbology was chosen largely as it mirrors that in Python. We'd absolutely be happy for the manuscript to be later typeset by ESurf like this...

Page 22, line 22. Near m/n, you may mention that this is the channel concavity. Perhaps, "m/n, i.e., channel concavity, $\approx$ 0.5. $\nu$ Added: "...interpreted from channel concavities of natural rivers at apparent topographic steady state."

Page 22, line 26. Remove "possible" (repeated later) $\nu$ Done

Page 23, line 2: Remove "performs as order-n, and as expected": redundant. You could keep "as expected" if you like. $\nu$ We disagree that this is redundant; we wish to demonstrate that the component is performing as it should in terms of its scaling. We've left this.

Page 23, lines 3-4: "Broadly linearly" sounds strange: two dimension terms, the first metaphorical. How about "approximately", or even just no modifier (it is so close to

linear)? ν Modified to "close-to-linearly"

Page 23, lines 17-19: "based on...2001b)." is a repeat of text at the start of section 5.2 (p. 22, l. 2-6). ν We've deleted this text the first time it appears.

Page 23, line 20: "storm depth" to "stormwater depth" (or something like this)? ν "...storm water depth..."

Page 23, lines 24-25: "looped upon" to "called within the loop", and remove "until...cease". If I understand you here, this is a simpler way of writing this. ν Now "in other words, the code block below the generator will repeat with fresh values for each iteration until the total time is elapsed, at which point the loop will cease"

Page 23, line 28: "and where it only rains" to "with rainfall occurring" for better parallel structure ν Done

Page 25, line 5: remove comma after "region" ν Done

Page 25, line 6: remove "using" after "by": unnecessary ν Done

Page 25, line 6: did you just write that stormwater depth is given by a gamma function? Also, "storm depth" should be turned into something like "stormwater depth" that makes more sense, per the above comment. ν Yeah, this was ambiguous as written. The durations are exponentially distributed, but the depths are gamma because they combine exponentials. The latter section is rephrased lightly to clarify. Changed also to "storm water depth".

Page 25, line 9: Does Bras' model incorporate latitude, integration over a day, etc.? And is it simple incident radiation (i.e. no diffuse or reflected)? I think that the latter is true from what you have written, but not sure about the former. ν This section has been expanded to address this explicitly: "The Radiation component calculates daily average extra-terrestrial and clear-sky shortwave radiation incident on a flat surface, based on latitude and day of the year (ASCE-EWRI, 2005). This component also calculates daily radiation ratio, defined as the ratio of cosine of solar angle of incidence for the true

sloped surface to that for a flat surface (Bras, 1990). The Radiation component does not explicitly calculate diffused and reflected radiation. The PotentialEvapotranspiration component uses the radiation ratio to calculate spatial net radiation using daily maximum and minimum temperature, and potential evapotranspiration (ASCE-EWRI, 2005; Zhou et al., 2013)." Note that we also had a reference manager issue with the reference that was formerly listed as "Snyder, 2005"; this should in fact have been rendered as "ASCE-EWRI, 2005" (a technical report chapter), as it now is.

Page 25, line 21: "and also" → "and" $\nu$ Done

Page 25, line 29: "shrubs cluster as they..." $\nu$ Much better! Done.

Page 26, line 12: "shallow" is ambiguous since you are writing about flow depth and slopes. I would write "low-slope" instead. Also, did you mean a different word than "urban"? $\nu$ Good call; did this. Also no, urban is right. The Bates algorithm was originally derived for use in city planning.

Page 26, line 29: "a uniform rainfall rate". Also, this is by filling the pits with water, I presume. $\nu$ I'm unclear what this comment is in reference to, sorry. The section around here seems OK to me, so I'm leaving it.

Page 28, line 4: You haven't mentioned visualization except for one reference to ParaView that you didn't follow up and a passing reference to matplotlib. Not actually a problem, but a note. $\nu$ This material was excluded mainly for reasons of manuscript length, and I think that's a decision we stand by. Lots more information about plotting and visualization can be found through the LL website.

Tables Table 1b: What about raster D8? $\nu$ This gets at an aspect of Landlab's architecture which we have specifically not covered in this paper, because it is still fluid in the code: although it is possible to build rasters and rectilinear grids with diagonal node connections in version 1, the details of this are not exposed to the user at this stage, and may yet be modified. Essentially, Landlab doesn't view "diagonals" as

[Figure]

true links (since they don't honour the graph theory rules that underlie Landlab's grid); any time they are invoked, they must be invoked separately from the true, orthogonal links. Since this functionality isn't user-facing, we've deliberately not covered it in this manuscript. I note that this was slightly ambiguous when we referred to "diagonal links" in table 2, so I've modified that description to now read "diagonal connections between nodes" (they aren't links!) This should make everything technically correct.

Table 2: Should rectilinear have rectangular cells, not "quasi-rectangular"? $\nu$ Not necessarily. This category also permits topologically warped rectangles (e.g., as in a global projection), should the user so desire.

Table 4: More of a comment: based on all boundaries being inactive, a 1D model in Landlab would have to have $3 \times (N + 2)$ nodes. So it really isn't optimized for 1D, it seems. $\nu$ Certainly not optimized, no, but still perfectly capable. Landlab doesn't actually perform component calculations on inactive links in most cases, so it's only really an issue of overhead in grid setup. During the iteration of a run, it would run pretty efficiently in most cases.

Table 5: A nice set of modules... but at risk of sounding ungrateful of the large amount of work that this was, it would be really great to see a depositional components for Landlab appear... in case this isn't obvious, this is not a criticism of the paper, but more a comment on where this work could lead. $\nu$ Yep yep, this is very much on our minds (see also comments to Campfort's review).

Figures

Figure 3: Please ask copyeditors to ensure that this appears on the same page as Table 1. $\nu$ Noted.

Figure 4: Are patches numbered in order of min(max(surrounding nodes or links))? (Question inspired by (b)(ii).) $\nu$ Nope. Patches are ordered by centroid position (i.e., by corner), according to the same scheme as all the other element types as described in

the main text. It's just a little unclear from this particular figure – which really focuses on the link directions – exactly where all the centroids are for these patches! I did in fact have to confirm this was right for myself, as it looks a little shonky.

Figure 7: Include line numbers. Also, flux → discharge, per our discussions? In addition, for this and the other figures with code, I am not sure if Âż> and ... are needed or are distracting... they are more the latter for me, I guess. But your decision in the end on style. $\nu$ In this case, we think "unit flux" is probably the right terminology, as it's a discharge per unit width of the face. We think we will retain the »>'s here and elsewhere, as this style explicitly echoes the "doctesting" style in Python that users will see widely through Landlab's documentation. We think it would be good to introduce them to it here, rather than adopt a different style here and another in the online documentation.

Figure 8: You use both "base level" and "baselevel"; check for consistency (2x/each). $\nu$ We've committed to "base level". We also noticed that we had inconsistencies between "time step" and "timestep", which have now been unified as "timestep".

Figure 9: Why not combine this with Figure 8, if they make the same plot? $\nu$ We think having them separate is less confusing, as they appear separately in the text, and it would be hard to explain in the caption alone why both were being shown.

Figure 10: Please tell us how many cells you computed over to help readers gauge their own compute times. $\nu$ I'm unsure how this is different to what we are already showing here. LL generally regards nodes as the primary element type, so the figure gives speeds for numbers of nodes. Given these are simply nxn grids, users could easily compute numbers of nodes for themselves from the information given (e.g., ncells = (sqrt(nnodes)-2.)**2 for a raster). In most of these cases, it makes basically no difference whether we count nodes or cells.

Figure 14: (b) Vertical axis: PFT is more precise than plant type? Although this is clear from everything else. Legend: shown in parts (b) and (c) of this figure. $\nu$ We've chosen to leave this, as it seemed very minor.

Supplement

No concerns.

Testing

During the review process, I took advantage of the fact that I am teaching a geomorphology class to test Landlab. In our modeling lab, the students, most of whom have little experience in programming, were all able to install and run Landlab. In some cases, differences between computers and Python versions caused a little confusion, but the code itself worked well. The students are currently working using it with a simple coupled channel–hillslope model to (1) find relationships between hillslope diffusivity, the stream power coefficient, and the drainage area of the hillslope–channel transition, (2) investigate which factors lead to changes in the time it takes to reach steady-state conditions in a landscape, and (3) explore the effects of changing grids, uplift patterns, stream-power exponents, and more on the model results. The overall reception was very positive in this mixed undergraduate–graduate class, with students realizing through the model how real landscapes form and evolve. The code that I used, modified from one of those supplied in the paper, is attached below. I intend to teach the lab again. One issue that I had while running this code on Linux within iPython is that the plotting tools would not release control of the shell to me (i.e. after plotting, I could not type anything more into the terminal, even after closing all of the graphical windows). This of course is not an issue to preclude publication, but I will add it as a ticket on GitHub.

$\nu$ Thanks for this feedback, it's very valuable to us. Please do submit a ticket, though past experience of plotting issues suggests to us that this will be an issue arising from matplotlib's idiosyncrasies rather than an actual issue with Landlab. Nonetheless, we'll take a look.

---

## Author Comment (AC4) · 24 Nov 2016

Dear Benjamin,

Many thanks for completing this review for us. As requested by the AE, please find below our replies to your comments. We've pasted each comment you made, then replied to it, and as necessary then copied in any modifications made in the text. We hope that these replies address your concerns to your satisfaction.

I also note that the process of pasting this text from Word to the ESurf online interface has resulted in our carefully formatted bullets now appearing as little inline "nu"

symbols. Sorry! Hope this isn't too annoying.

Thanks again,

Dan (on behalf of all the authors)

Review of Hobley et al. creative computing with Landlab

With Landlab, Hobley et al. present a refreshing framework to integrate numerical solutions for planetary surface processes. The model is fully open source and targets both end-users and developers providing an extended library of model structures and implementation strategies. The wealth in existing models and model structures makes the paper timely for the geophysical earth surface community and I recommend the paper for publication after consideration of some remarks.

Before reading the paper and because of its intention: i.e. providing an integrative numerical modeling toolkit, I formulated some questions which I would like to see addressed in such a paper. In the following I list those questions and discuss to what extend I found satisfying answers and where the paper can be improved.

* General ease of use of the numerical model. I congratulate the authors for their well-functioning and carefully documented source codes. $\nu$ Thanks!

* Debugging tools I highly value the availability of unit tests to evaluate the model code. This is one of the aspects in which Landlab differentiates from other numerical earth surface models. $\nu$ Again, thanks. Good to be reassured that effort on this front has been worth it...!

* Input/output data structures. Compatibility with other tools available. Well implemented and properly discussed in the paper $\nu$ Thanks.

* Discretization of the continuous solution to grids. Shifting between different kind of grids. Another major contribution of the paper is the way in which different discretization schemes and the numerical grids which come with them are presented. Nonetheless, while reading section 3.1, I sometimes had the feeling reading technical notes rather than a paper on earth surface dynamics. I propose that the authors discuss, somewhere in this section, the importance of different grid implementations and its relevance for the field of earth surface dynamics. E.g. the use of structured grids has major implications in terms of landscape symmetry as elegantly documented by Braun and Sambridge (1997). It would also be interesting to briefly discuss the implications of using raster versus voronoi grids and to what extent the model supports conversion between the methods. A first example which comes to mind in the framework of LEMs (which is only partly what the model is designed for) is the simulation of tectonic shortening which is often executed on irregular grids (Willett, 1999), whereas other landscape processes such as nonlinear diffusion (Perron, 2011) might benefit from structured grids for computational performance. Does the grid structure of the model allows for grid refinement (e.g. Künze and Lunati, 2012)? $\nu$ The first of these points is a good idea, but we feel we've already done this fairly well in the manuscript. The paragraph now at p. 8 lns 17-24 covers this ground at a level we already thought was commensurate with the scale of this paper, and indeed does already briefly touch on Braun & Sambridge's insights. Nonetheless, we have added an additional clause with a little more detail from Braun & Sambridge ("Irregular grids avoid some of the cardinal direction artifacts than can form on regular grids, such as linear networks and linear drainage divides, as well as consequent biases in measured channel metrics like drainage density, river length, and channel slope (Braun & Sambridge, 1997).") Regarding the later comments here, however, we are reluctant to expand too much on what we have regarding explicit conversion between grid types. Landlab is not, and is not really at this time ever intended to be, a GIS tool. Operations like interpolation between grids are better accomplished in other, more specialised software in our view. At the moment, in part for similar reasons and also simply because of the lack of a pressing application, we do not allow grid refinement or densification. Past experience of some of the Landlab team (GT, NG) with CHILD – which does support this kind of ad-hoc grid modification in some instances – suggests that this kind of process can

**ESurfD**
be extremely challenging to implement. This would only be more so in Landlab, where densification would need to be accommodated not only within the grid, but its consequences also propagated out into many of the components. Nonetheless, we do intend that future versions of Landab may explicitly support grid deformation, if not refinement per se. That all said, we should probably also note that specifically regarding comparison between results obtained from different grid types, this is an active field of research for the Landlab team (see, e.g., Gasparini et al., Eos Trans AGU 2014, EP51E-3564), and future publications will address this issue specifically.

* Parallelization and suitability for supercomputing One essential asset of a robust model structure, especially if the authors target a large user community, is the possibility to port the source code to a larger computer infrastructure. This item is currently not covered in the paper and it would be good to discuss on the potential of Landlab to be parallelized and the limitations which comes with it. In case the model supports parallelization, how do you take care for changing drainage areas while executing the model on different computational blocks? $\nu$ Again, this material was not present in the paper primarily for reasons of space in what is already a long manuscript. As you are of course alluding to, parallelizing a heavily componentized software architecture is significantly more challenging than a fully compiled and stable code, as the programmer can not necessarily know ahead of time the sequencing of the calls to various parts of the code. This more or less rules out the idea of "parallelizing Landlab" in the broadest sense. However, parallelization is much more possible within individual components – and is implemented in Flexure, as proof of concept. However, given your explicit interest, we've added a short additional section:

"3.3.3 Parallelization

Together, the componentized nature of Landlab and the level of flexibility afforded to the user conspire to rule out the idea of Landlab as a whole being highly optimized through parallelization. However, there is great potential for parallelization of Landlab at the component level, since the run methods of each component are entirely selfcontained. As proof of concept, the Flexure component has already been parallelized (see online code and documentation). Although in Landlab version 1.0 we have not had a compelling enough use case to invest significant time in such work, many of the components already in the library would be amenable to parallelization in this style, and this could be done in future releases."

* 3 spatial dimensions Many earth surface processes require a 3-dimensional spatial discretization. Amongst others: hydrological processes, ice dynamics, terrestrial heat advection and diffusion, and soil dynamics. As the authors explicitly state that Landlab "is not a landscape evolution model ...rather, it presents a framework under which a wide variety of models can be implemented using it tools, including hydrologic ...", I am a little surprised that the use of a third spatial dimension is never mentioned in the text. I do not suggest that the 3rd dimension should be included in the current release of the software but I would find it interesting to see a discussion on the possibility of adding a third spatial dimension to the model. $\nu$ I'm assuming here that you are simply advocating that we discuss whether Landlab be able to run three dimensional grids. I don't feel this is necessarily close enough to the remit of Landlab (which is at the moment, fundamentally, a two-dimensional framework), as we set it out in the introduction, to be worth spending time on in the manuscript.

* Numerical accuracy. During last decades, many numerical models have been developed mainly focusing on (i) earth surface processes being simulated or (ii) the performance of the numerical simulations. Much less attention has been given to the numerical accuracy of the developed models. This issue is also not very well covered in this manuscript. Nonetheless, recent work has shown that numerical accuracy significantly influences model performance, not only when topographical knickpoints are present but also when lateral displacement of topography needs to be accounted for (Campforts et al., 2016; Campforts and Govers, 2015). Moreover, it would be good that Landlab also offers some analytical solutions for the numerical processes being simulated or that the authors discuss how this could be achieved as analytical solu-
tions are the benchmark against which different solution strategies can be evaluated. As such, a benchmark solution would be available which cannot only serve as a tool to evaluate the accuracy of the current model structure but is also of aid to new users in order to test new models and numerical solutions. For example, an analytical solution for the stream power model can be quite easily found using the slope patch solution of Royden and Perron (2013) but also for numerical solutions not directly related to landscape evolution, analytical solutions do exist (Stüwe et al., 1994). Numerical accuracy is sometimes mentioned throughout the text (e.g. when discussing the influence of timesteps on line 30, p. 15) but it would be good to elaborate on this point and cover this relevant issue in more detail $\nu$ As you note, this manuscript does not dwell on issues of computational accuracy. However, this is mainly because this text is focused on the functionality of Landlab as a whole, whereas issues of numerical accuracy are relevant to each component individually. On this basis, we would prefer not to greatly expand our coverage of this issue; rather, it will be discussed as relevant in future publications which are focused on each novel component in turn. For instance, Adams et al., in review, doi: 10.5194/gmd-2016-277 does this in some detail for the implementation of the overland flow model in section 5.4; a manuscript in prep from Siddhartha Nudurupati on the vegetation CA seen in section 5.3 does something similar for that component. [I should also note as an aside that the Adams et al. manuscript is now citeable in review at GMD, so we have reinstated references to it in the text as appropriate.] Likewise, the diffusional and stream power models illustrated in sections 5.1 and 5.2 are simply reimplementations of algorithms already comprehensively described by other authors; existing publications describing those schemes specifically (c.f. Braun & Willett, 2013 for stream power; e.g. Slingerland & Kump, 2011, Chapter 5 for discussion of central differencing schemes for diffusion) already provide much of this information, and we assert that it is not necessary to repeat here.

* To what extend is the model oriented towards integration of measured data into the numerical model? Major advances can be made in the field of earth surface processes by combining numerical models with field data (Fox et al., 2015; Glotzbach, 2015). I

**ESurfD**
think the framework presented in this paper offers an excellent opportunity for such an approach and could be of high value for otherwise more data oriented researchers. It would be interesting to see some explicit suggestions or remarks on how to combine the Landlab framework with field data. $\nu$ In the interests of space and keeping the manuscript focused, we chose not to include this material. We should also re-emphasise that this kind of functionality will probably appear at the component level, so would be best addressed in future, component-level publications. We anticipate that there will be a good number of these, and indeed, forthcoming publications from the core team (e.g., Siddhartha Nudurupati et al., in prep; Hobley et al., in prep) are already making strides in this direction.

* Particle tracking One very interesting aspect of numerical models is to trace back particles through time. I am wondering whether Landlab allows fingerprinting of for example sediments. Is it possible to track a sediment particle all the way from the hillslopes to the outlet of the drainage network? Can this be done for all particles and how feasible is that in terms of memory allocation and data storage? Particle tracking would offer great potential for the model to be further combined with e.g. detrital dating methods (thermochron data (Herman et al., 2015)/ CRN data (Mudd, 2016)) ). $\nu$ This is indeed an interesting aspect, and we very much intend for future components to include a particle tracker. However, given that (as Wickert also noted) the current suite of components at version 1.0 is not particularly sediment- or deposition-orientated, we haven't prioritized this functionality, and it seems unnecessary to speculate on it here without having the work in hand.

* An interoperability interface

In the abstract, Landlab is presented as a modular framework with a strong interoperability. The paper comes up to this promise and presents four contrasting model designs. I enjoyed reading these sections and I think they strongly contribute to the illustrative nature of the paper. Nonetheless, I think this section could be even stronger if the authors also point out the advantage having the different model approaches under the roof of one single model infrastructure. In my opinion, one of the main assets of Landlab is not necessarily the capacity to execute separate model designs but exactly that they can be combined. For example, until now, little attention has been given to the role of vegetation in landscape evolution models. Nonetheless, it has been shown that vegetation might have a very strong influence on landscape evolution (Collins, 2004). The cellular automaton presented in sections 5.3 could therefore be neatly integrated with section 5.1 and/or 5.2. Likewise, hydrological processes are shown to govern stochastic processes like shallow landsliding (Montgomery and Dietrich, 1994). Hence section 5.4 can be perfectly linked up with the other sections. I would keep the sections in their current form as they are very illustrative of the model performance but it would be definitely nice to discuss and eventually illustrate the potential to integrate all or some of these modules. $\nu$ We had hoped that this point was coming across strongly enough already, but we're happy to address this a bit more forcefully. We've added "We hope that these examples will also serve as an illustration of the potential power of the Landlab framework to enable novel or under-explored process interaction studies (e.g., of vegetation on landscape evolution; of surface hydrology on stochastic surface processes)". . . to the introduction of section 5.

Minor comments

Overall the paper is very fluently written and I enjoyed reading it. I endorse the comments of W. Schwanghart and A Wickert. In addition, I have a few minor comments: Page 2-5: Introduction. I get the point the authors try to make but the way the introduction is written feels like the authors want to promote their product as the one and single framework to be used for future model development. Although I agree with the authors that the structure they provide can offer excellent guidance for new model development and the 'gridding engine' can definitely save many hours of numerical coding headache, I am not so sure whether I agree it is a good idea to demotivate building code from scratch. The latter is still the best way to become familiar with the implications and limitations which come with numerical software design and to realize to power and caveats of numerical models. I second W. Schwanghart's comment in suggesting that many open source initiatives exist, although sometimes written in commercial software (e.g. TopoToolbox and the LEM TTLEM is fully open access; SIGNUM (Refice et al., 2012)) or even in Python, advanced GIS packages are available (LSDTopoTools https://github.com/LSDtopotools). $\nu$ We aren't sure what exactly we can change here (outwith our modifications pertinent to Wolfgang's comment). It is certainly not our intention to demotivate future readers to code up stuff for themselves! That said, it's clearly the purpose of this manuscript to set out Landlab's shop stall. We've made a couple of minor tweaks to the language used to emphasise more that the issues described here pertain specifically to issues for professional research scientists who just want to produce high quality software, and don't necessarily carry over to considerations around learning or skill acquisition (See track changes in paragraph 2, p. 2). Hopefully a future publication in a different forum will expound on Landlab's advantages as a pedagogical environment.

Page 10, line 24: Analytical method. I would use a different wording as analytical typically refers to the analytical solution of e.g. PDE's. What about grid methods? $\nu$ Good call on changing this. We've selected "Computational methods".

Page 15-16, line 23-5: I find this very interesting. Does the model allow for constraints on the timestep in the case that implicit methods are being used? Although they are indeed unconditionally stable, the use of large (main model) timesteps in LEMs might result in in very sudden topographical changes causing the presence of artificial steps in the landscape. E.g using an uplift rate of 1mm/yr in combination with timesteps of 10ky results in a sudden uplift of 10m in the LEM which initiates artificial knickpoints at the baselevel of rivers. $\nu$ Aha, good question. As is implied (though I don't think outright stated) in this section and its associated figure, when Landlab has a truly or at least largely implicit solution, it does _not_ seek to impose any internal restrictions on that timestep. As you rightly note, this runs the risk that the user can take inappropriately long timesteps. We agonised about this, but decided that in a very flexible environment

like Landlab, adding arbitrary timestep restrictions on implicit methods would create more problems and confusion than it would solve. In such cases, the component documentation normally makes reference to this issue, and the user is encouraged to think about it for themselves – the "appropriate timestep" would be very much situation dependent, so we can't really legislate for it in a general way. We've added a sentence here to make explicit for the reader that this is indeed "a thing": "Note also that where components employ implicit solutions, there may be no internal limit to the timestep at all (e.g., Braun and Willett's (2013) Fastscape algorithms for stream power). In such cases, Landlab will make no check on the imposed timestep, and the user must ensure that the imposed dt is appropriate under the boundary and initial conditions that they are running. For instance, the Braun-Willett algorithm ceases to behave in a truly timestep-independent fashion under transient conditions, but in a way that still permits timesteps larger than would be imposed under an explicit Courant condition (for more details see their Appendix B). However, those authors did not propose an alternative scheme to limit the timestep in such cases, and consequently Landlab also does not. A user of this component is assumed to have read the component documentation and taken on board that this is potentially an issue, and to have taken steps to check that their output is behaving sensibly and is not highly sensitive to changes in the supplied timestep. We reiterate that it is ultimately the user's responsibility to check that the provided dt is appropriate to the modelling scenario in hand."

Page 18-19: nice illustration on the use of finite volume methods.

Page 22, line 31: what about maximum timesteps for implicit schemes? See comment above (Page 15-16) $\nu$ Now addressed explicitly in that section.

Page 24 line 22: I would get rid of the sentence starting with Figure 13. I also do not see the additional value of Fig 13 for this paper. $\nu$ We would prefer to retain the figure, as it neatly encapsulates a key functionality of Landlab that isn't well illustrated in any of the other figures.

Figures

Great figures in general.

Figure 1: Nice images but I find it a missed opportunity to illustrate the broad range of problems Landlab itself is capable to simulate. $\nu$ In principle, Landlab would be able to simulate any and all of these processes, given a motivated user. In most of these cases, moves are already afoot to add these functions into Landlab for a future release. . .

Not sure what the additional value of the code snippets is. Personally, I find the Landlab wiki page much more insightful in this perspective. $\nu$ While we definitely would want a user to look at the wiki, we wanted this paper to be able to stand alone, without forcing a reader to go online to get more resources. Hence we have included the code snippets in the text.

Testing:

I am not very familiar with python and I second the comments of referee 1 who did elaborate testing with a large user group. Nevertheless, I installed the software and executed some of the tutorials which I found very easy to understand and clearly documented. I had no single problem when executing the code. A very small suggestion would be to add a link to the 'coupled_params_storms.txt' file in the 'Getting to know the Landlab component library' tutorial on https://nbviewer.jupyter.org/github/landlab/tutorials/blob/master/component_tutorial/component_tutorial.ipynb, similar to the link given for the 'coupled_params.txt' file. $\nu$ Thanks for the recommendation. I'll add it as a ticket on Github.